# A fast-charging/discharging and long-term stable artificial electrode enabled by space charge storage mechanism

Linyi Zhao[1,4], Tiansheng Wang[1,4], Fengkai Zuo[1], Zhengyu Ju[2], Yuhao Li[1], Qiang Li[1], Yue Zhu [2,3] ✉, Hongsen Li [1] ✉ & Guihua Yu [2] ✉

Lithium-ion batteries with fast-charging/discharging properties are urgently needed for the mass adoption of electric vehicles. Here, we show that fast charging/discharging, long-term stable and high energy charge-storage properties can be realized in an artificial electrode made from a mixed electronic/ionic conductor material (Fe/Li$_x$M, where M = O, F, S, N) enabled by a space charge principle. Particularly, the Fe/Li$_2$O electrode is able to be charged/discharged to 126 mAh g$^{-1}$ in 6 s at a high current density of up to 50 A g$^{-1}$, and it also shows stable cycling performance for 30,000 cycles at a current density of 10 A g$^{-1}$, with a mass-loading of ~2.5 mg cm$^{-2}$ of the electrode materials. This study demonstrates the critical role of the space charge storage mechanism in advancing electrochemical energy storage and provides an unconventional perspective for designing high-performance anode materials for lithium-ion batteries.

Fast-charging/discharging batteries are a crucial power component to allow faster and farther travel, advancing the public adoption of future electric vehicles (EVs)[1–3]. Developing high-rate anode materials is one of the kernels of realizing fast-charging/discharging lithium-ion batteries (LIBs), which currently prevail in the commercial market[4,5]. Although commercial graphite anodes and silicon-carbon anodes have been widely used and studied due to their high theoretical capacities and low standard electrode potentials vs. lithium, they fail to deliver satisfactory capacities when charged and discharged at high rates[6–9]. In addition, slow interfacial dynamics and large overpotential inhomogeneity (polarization) under high-rate charge–discharge conditions in these anodes lead to uncontrolled deposition of lithium dendrites on the electrode surface, which may cause short circuits and thermal runaway[10–13]. Therefore, balancing between the high energy density and fast charging/discharging performance of LIBs is still very challenging.

Up to date, existing strategies of developing fast-charging/discharging anode materials mainly rely on composition/morphology optimization and electrode-level structural design, yet limited effort has been put into escaping from traditional energy storage mechanisms such as intercalation–deintercalation and conversion[14–17]. Conventionally, upon lithium insertion into an intercalation electrode, lithium ions, and electrons are stored simultaneously within a single host, where lithium ions occupy interstitial sites provided by the crystal, accompanied by accommodation of electrons at redox centers (mostly transition metals)[18–20]. When the lithium-inserted phase is not thermodynamically stable, decomposition (conversion) occurs, resulting in a composite product. What is intriguing is that a special storage mechanism exists in this end product and also in certain rationally fabricated composite materials[21,22]. For instance, charge storage in some electrochemically in-situ generated mixed conductor systems relies on an intimately contacting interface (space charge storage mechanism), whereby lithium ions are stored on the ionic conductor side of the contact and electrons on the electronic conductor side[23–26]. This phenomenon has been reported in the context of understanding extra storage capacity in a series of conversion-type transition metal compounds[27,28]. Because in this storage mode, charge storage is decoupled, the greatest advantage of this mechanism is that it can attain very high power density, and if the effective storage area is not

[1]College of Physics, Qingdao University, Qingdao 266071, China. [2]Materials Science and Engineering Program and Walker Department of Mechanical Engineering, The University of Texas at Austin, Austin, TX 78712, USA. [3]School of Materials Science and Engineering, Ocean University of China, Qingdao 266404, China. [4]These authors contributed equally: Linyi Zhao, Tiansheng Wang. ✉e-mail: zhuyue@ouc.edu.cn; hsli@qdu.edu.cn; ghyu@austin.utexas.edu

sacrificed, also high-energy density, while stable long-term performance can be maintained due to the nature of a pure interfacial process[29–31].

Herein, we report a type of artificially designed electrodes employing a novel high-performance mixed electronic/ionic conductor material that is based predominantly on the space charge storage mechanism. A series of interfacially optimized electrode materials (Fe/Li$_x$M where M = O, F, S, N) with Fe/Li$_2$O as a representative have been successfully constructed through a facile lithium thermal displacement reaction, showing excellent electrochemical performance in several key aspects. At a high charging/discharging current density of 50 A g$^{-1}$, the Fe/Li$_2$O electrode retains 126 mAh g$^{-1}$ and sustains 30,000 cycles with negligible capacity loss at the charging/discharging current density of 10 A g$^{-1}$, rivaling many fast-charging electrochemical storage electrodes reported in the literature. The energy storage in the Fe/Li$_2$O electrode is verified to be occurring mainly at the designed interface, ensuring decoupled and rapid charge transport that is not available in conventional electrode materials. Thermodynamic fitting and magnetic analysis, combined with structural characterization, prove the presence of only limited conversion reaction due to reactant spatial distribution confinement even at low rates (current density of 1 A g$^{-1}$), which otherwise will overshadow the desired space charge storage mechanism. Bringing this superior mechanism into designing faster charging electrode materials creates a viable approach to bridge the performance gap between current specifications and increasing requirements for future battery applications.

## Results and discussion
### Design principle and material characterization
The essence of this work is to design electrode materials that harness mixed conductor heterogeneous interfaces for charge storage to enable exceptionally high-power densities in batteries. The schematics in Fig. 1a, b compare the homogenous charge storage in conventional materials with the space charge storage mechanism that could contribute to markedly high chemical diffusions of both ions and electrons in each conducting phase (dashed line arrows). In a normal bulk storage mechanism, a material simultaneously accommodates ions and electrons and must sustain both high electronic and ionic conductivities in order to achieve a high power density (Fig. 1a). While in a space charge storage mechanism, a material combines different phases that separately store and transport ions and electrons in its individual space charge zones (Fig. 1b). In this context, fabricating a mixed ionic-electronic conducting material is the key to realize the space charge storage mechanism. For our chosen material system, this mixed conductor is an artificial composite synthesized by lithium thermal reduction reaction, as depicted in Fig. 1c, d. The syntheses were carried out in an inert environment, with tantalum used as the reaction vessel to prevent high temperatures and strong reduction effects caused by molten alkali metals[32–34], and the final products needed to be stored in a strictly air-isolated environment. Taking Fe/Li$_2$O as an example, the X-ray diffraction (XRD) pattern of the as-synthesized product is shown in Fig. 1e, confirming that all the diffraction peaks can be indexed to the PDF cards of elemental Fe (JCPDS No.06-0696) and Li$_2$O (JCPDS No.12-0254). Field emission scanning electron microscopy (FESEM) image inset clearly displays its spherical morphology, constituted by an even smaller microscopic structure. The ferromagnetic characteristics signal is illustrated by the M–H curve of the Fe/Li$_2$O at room temperature in Fig. 1f. The saturation magnetization of the material and the nanoparticle size can be estimated by fitting with the modified Langevin function[35,36]:

$$M = M_0 L\left(\frac{\mu_p H}{kT}\right) + \chi_a H \tag{1}$$

where $L(x) = \coth x - \frac{1}{x}$, and $\mu_p$ is the magnetic moment of a single particle. Considering the bcc unit cell of Fe with a lattice constant of

0.287 nm, each unit cell contains two Fe atoms, the best-fit particle radius is $R = 4.11$ nm, and the saturation magnetization intensity is 88.8 emu g$^{-1}$. According to previous studies, the magnetization of pure Fe nanoparticles with the same size is 210 emu g$^{-1}$[37,38], thus the calculated content of Fe is approximately 42% through the comparison calculation. This is consistent with a more direct measurement using inductively coupled plasma (ICP) technique, and the results are given in Supplementary Table S1. Supplementary Fig. S1 shows the high-resolution X-ray photoelectron spectroscopy (XPS) scanning measurement spectra of the product, indicating that it is composed of three elements, Fe, O, and Li, and the valence states of these three elements are Fe$^0$, O$^{2-}$ and Li$^+$, respectively. The above characterization results clearly indicate that the as-prepared Fe/Li$_2$O is a pure phase material in terms of its chemical composition and basic magnetic properties. Considering that the material can be regarded as a mixed ionic–electronic conductor at the macroscopic scale, we have also characterized the electronic and ionic conductivities of the as-prepared Fe/Li$_2$O. For detailed information, please refer to Supplementary Note 1 and Supplementary Table S2.

### Electrochemical performance
The electrochemical performance of the Fe/Li$_2$O as an anode electrode in LIBs was first evaluated in the form of coin cells. As shown in Fig. 2a, it can withstand a high current density of 50 A g$^{-1}$ and is able to deliver a capacity of 126 mAh g$^{-1}$, meaning that the minimal time required for a complete charge/discharge is only 6 s. The Fe/Li$_2$O electrode also exhibits outstanding rate performance at other current densities. When the current density is ramped down from 50 to 40, 30, 20, 10, 5, 2.5, and 1 A g$^{-1}$, the specific capacities are as high as 126, 145, 165, 200, 260, 306, 336, and 407 mAh g$^{-1}$, respectively (Fig. 2b). Figure 2c displays a representative cyclic voltammetry (CV) curve of the Fe/Li$_2$O electrode at a scan rate of 10 mV s$^{-1}$. Obviously, it exhibits a capacitive characteristic with broad peaks, indicating a predominantly non-diffusion controlled charge storage mechanism[39,40]. Figure 2d shows the long-term cycling performance of the electrode at a high current density of 10 A g$^{-1}$. Clearly, the Fe/Li$_2$O electrode exhibits excellent cycling durability and it can cycle reliably for 30,000 cycles without any capacity decay. Such excellent electrochemical performance places the current Fe/Li$_2$O favorably among many other fast-charging/discharging materials, including earlier studies employing a similar storage mechanism (Fig. 2e)[41–47]. When comparing parameters such as capacity retention (CR), high-rate specific capacity (SCH), low-rate specific capacity (SCL), maximum rate capability (Rat.), maximum cycle number (cycle ability, Cyc.), and final capacity in long cycles (FC), the Fe/Li$_2$O electrode is significantly superior in most sections. Figure 2f further compares the rate performance with previously reported fast-charging electrodes using comparable mass loadings (Supplementary Table S4)[41–45]. The electrochemical properties of the designed Fe/Li$_2$O electrode are exceedingly better than other listed materials reported previously. In addition, to evaluate the practical capability of the designed Fe/Li$_2$O anode, a coin-type Fe/Li$_2$O||LiFePO$_4$ full cell was fabricated and examined. For detailed relevant information and the performance of the full cell, please refer to Supplementary Note 2 and Supplementary Fig. S2. Furthermore, we compared the energy density and power density of the Fe/Li$_2$O electrode and the Fe/Li$_2$O||LiFePO$_4$ full cell with representative previously published results on fast-charging materials and devices. Both the electrode and the full cell based on the Fe/Li$_2$O material exhibit clear power density advantages. The detailed data, including the calculation method for power density, are given in Supplementary Note 3 and Supplementary Fig. S3.

### Comprehensive analysis of the charge storage mechanism
In order to reveal the fundamental basis of such electrochemical performance and to verify whether the proposed space charge storage is achieved in the constructed mixed electronic/ionic conductor

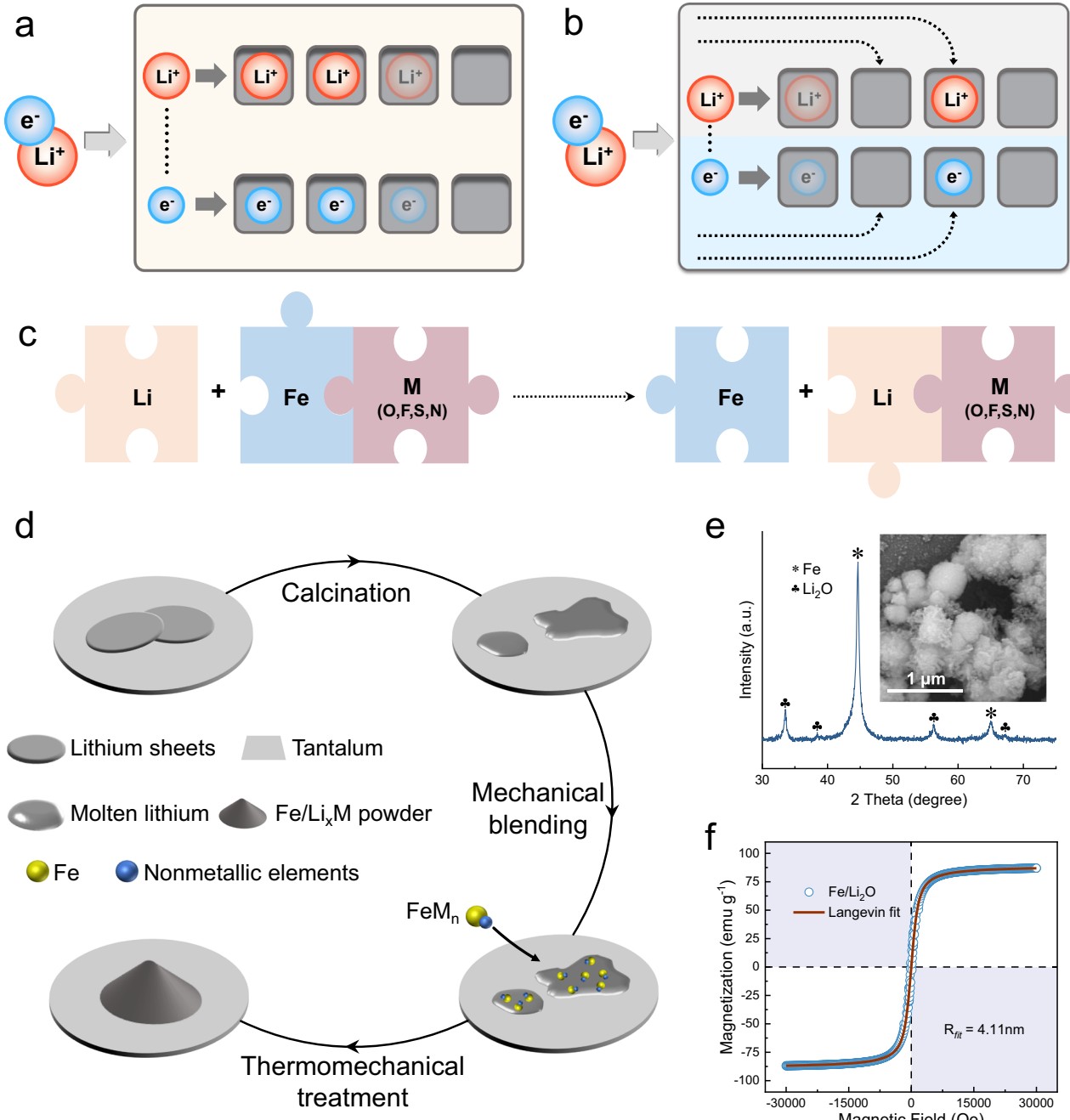

**Fig. 1 | Design principles and characterization of the Fe/Li₂O active material.**
**a**, **b** Schematics of the different chemical diffusion mechanisms (bulk and space charge storage). **c** Pathway of the lithium thermal displacement reaction.
**d** Schematic diagram of the preparation process of Fe/Li$_x$M (M = O, F, S, N). **e** XRD pattern of the as-synthesized Fe/Li₂O. Inset: FESEM image of the Fe/Li₂O. The scale bar is 1 μm. **f** M–H curve of the Fe/Li₂O at room temperature and the corresponding Langevin fitting curve.

interface, we carried out further characterization of the Fe/Li₂O material. It is well known from previous battery related reports on iron oxide materials that FeO$_x$ undergoes a conversion from FeO$_x$ to a mixed composite of Fe/Li₂O at low potentials vs lithium[48,49]. During a reversible process of increasing potential, the electrochemically generated Fe/Li₂O is normally oxidized back to FeO[27]. To elucidate the elemental valence state changes during the electrochemical reactions of the designed Fe/Li₂O electrode, XPS was employed to probe the sample at selective stages of the process. Figure 3a shows XPS spectra of the Fe element upon discharging the Fe/Li₂O electrode to 0.01 V and charging back to 3 V at 1 A g⁻¹. Compared to that of the initial as-synthesized material (Supplementary Fig. S1a), the overall valence state of Fe remains mostly unchanged and maintains a zero-valent state upon discharging to 0.01 V. After being fully charged to 3 V, most of the Fe remains in the same zero-valent state, but a small portion of it could have been oxidized to Fe²⁺. This is similar to the conversion reactions that occur during the charging process of conventional conversion-type iron oxide or related anode materials (such as Fe₃O₄), but differs only in the degree of oxidation. Therefore, in this case, the re-oxidation conversion extent of the Fe/Li₂O is significantly lower, and it could also be seen from the XRD measurements on the electrode when discharged to 0.01 V and charged to 3 V, as shown in Supplementary Fig. S4. It is interesting and unexpected that no characteristic peaks of iron-containing compound substances such as FeO were

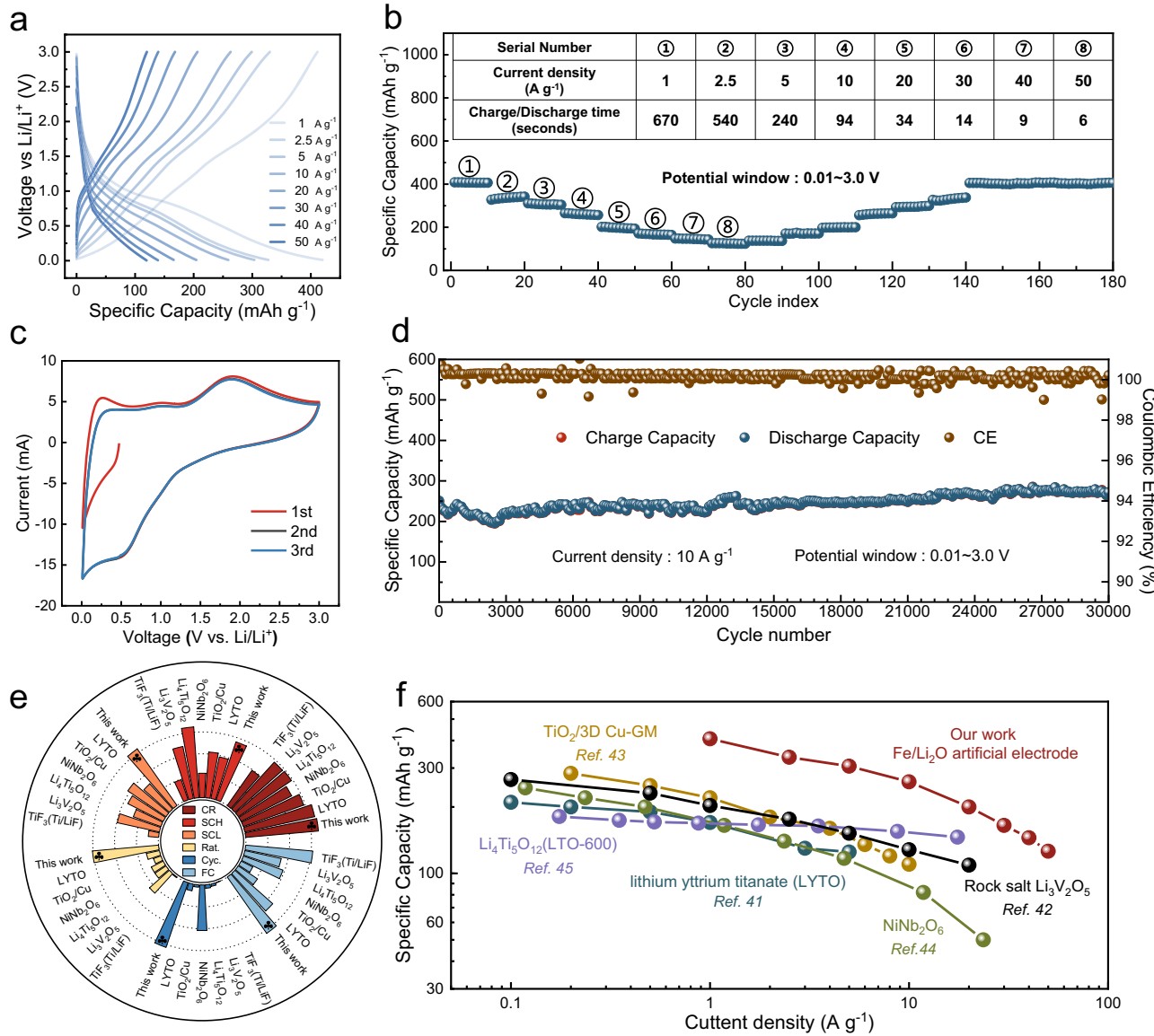

**Fig. 2 | Electrochemical performance of the Fe/Li₂O in LIBs. a** Charge–discharge curves of the Fe/Li₂O electrode at different current densities. **b** Rate performance of the Fe/Li₂O electrode. **c** CV curve of the Fe/Li₂O with a scan rate of 10 mV s⁻¹. **d** Long-term cycling of the Fe/Li₂O electrode at current density of 10 A g⁻¹. **e** Overall performance comparison of the Fe/Li₂O electrode with relevant reports of fast charging materials in CR (%), SCH (mAh g⁻¹), SCL (mAh g⁻¹), Rat. (C), Cyc. (number), FC (mAh g⁻¹). The detailed information of these materials in terms of such 6 key aspects are provided in Supplementary Table S3. **f** Capacity–current density performance comparison of the Fe/Li₂O electrode with relevant reports of fast charging materials.

observed in the XRD patterns. Regardless of the terminating voltage, only the characteristic peaks of Fe and Li₂O (excluding the Cu current collector) can be observed. This is somewhat contradictory with the characterization results obtained from the XPS at the fully charged state. Based on the characteristics and limitations of XRD as a characterization technique, we speculate that the reasons for the failure to observe $Fe^{2+}$ in XRD may include low content, ultrasmall particle size, amorphous structure of the products, or other factors[50–52].

To resolve this inconsistency and to overcome the limitations of XRD, high-resolution transmission electron microscopy (HRTEM) characterization of an electrode made from commercial Fe₃O₄ (Fig. 3b–d) and the Fe/Li₂O electrode (Fig. 3e–g) at different charge and discharge states was conducted, and the former serves the purpose of comparison. It is evident that the Fe/Li₂O composite formed in the commercial Fe₃O₄ electrode (Fig. 3c) upon discharging to low potentials is different from that synthesized via lithium thermal reduction, which exhibits a more pronounced spatial aggregation state

in terms of the distribution of Fe or Li₂O nanoparticles (Fig. 3e). Interestingly, this spatial aggregation state can be well preserved upon full discharge (Fig. 3f). Upon charging to a higher potential (3 V), the Fe/Li₂O formed through electrochemical transformation, namely the reduction of commercial Fe₃O₄, tends to undergo more thorough and uniform oxidation reactions, leading to most of the Fe and Li₂O being converted to form FeO (Fig. 3d). In contrast, the chemically synthesized Fe/Li₂O with certain distinct spatial distribution characteristics tends to exhibit conversion only at the boundaries between the aggregated regions. The majority of the Fe and Li₂O nanoparticles remain unchanged in their original states (Fig. 3g), and the sparsely formed FeO in this sample could explain the absence of its characteristic peaks in the XRD measurement. Furthermore, in order to check the stability of Fe and Li₂O in terms of their particle size after cycling, we first acquired the M-H curve of the Fe/Li₂O electrode and the corresponding Langevin fitting result after cycling for 200 cycles at a current density of 1 A g⁻¹ (Supplementary Fig. S5a). The Langevin

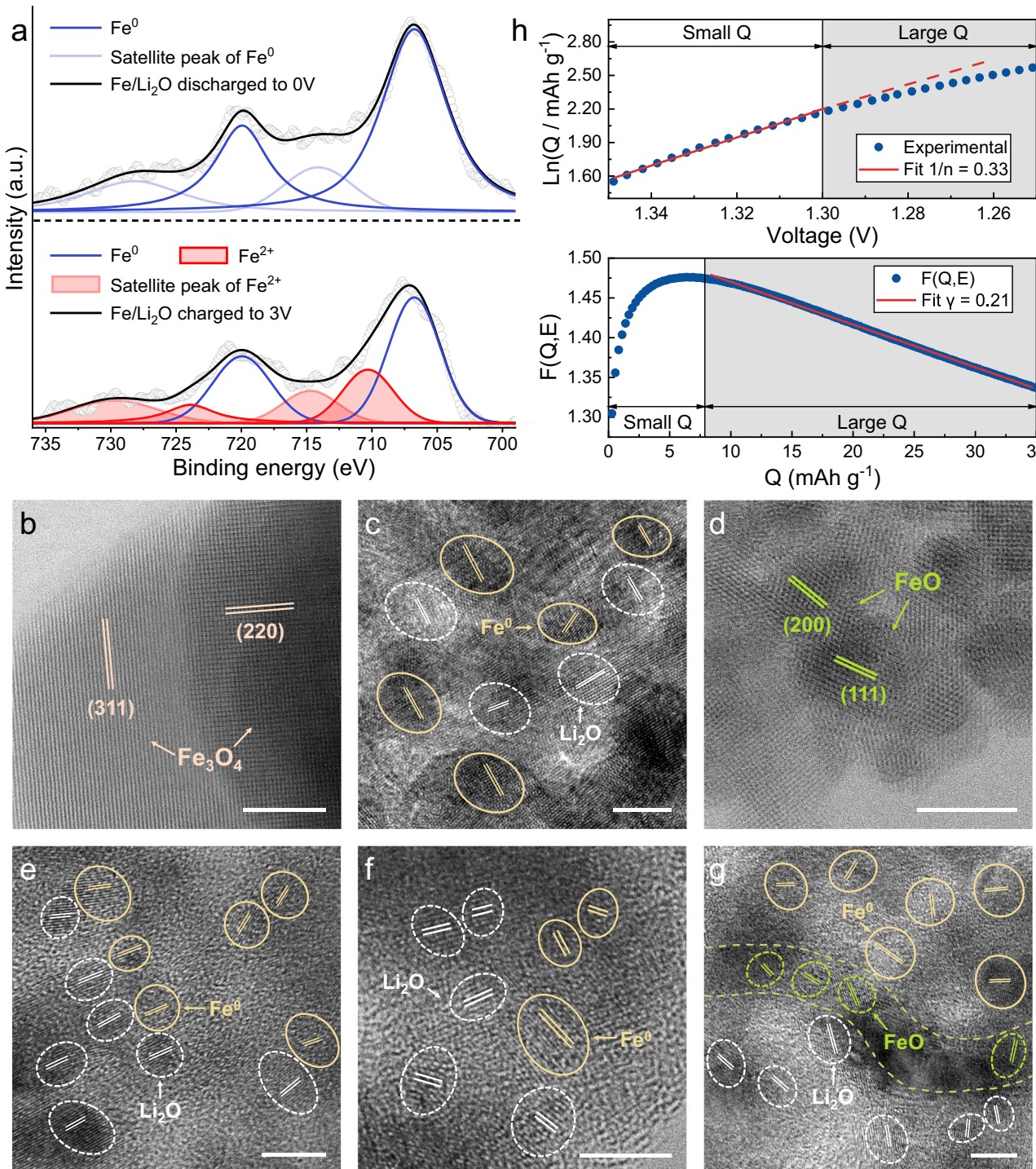

**Fig. 3 | Chemical composition and components spatial distribution evolution of the Fe/Li$_2$O electrodes. a** High-resolution XPS spectra of the Fe element in the Fe/Li$_2$O electrode at different charge and discharge potentials. **b**–**d** HRTEM images of the commercial Fe$_3$O$_4$ in **b** (initial state), **c** (discharged to 0.01 V), and **d** (charged back to 3 V). **e**–**g** HRTEM images of the Fe/Li$_2$O in **e** (initial state), **f** (discharged to 0.01 V), and **g** (charged back to 3 V). The scale bar is 5 nm. **h** Ln(Q)–V fitting curve and F(Q,E)–Q fitting curve of the Fe/Li$_2$O electrode.

fitting gives a particle radius of $R = 3.88$ nm and a saturation magnetization intensity of 89.5 emu g$^{-1}$ after 200 cycles (compared to the initial particle size of $R = 4.11$ nm and the saturation magnetization intensity of 88.8 emu g$^{-1}$ before cycling, as shown in Fig.1f). Next from the HRTEM image (Supplementary Fig. S5b), it can be observed that the average particle size of Li$_2$O in the electrode is approximately 4.90 nm after 200 cycles (compared to an average particle size of about 4.22 nm before cycling, as shown in Fig.3e). From these results, it

can be concluded that the particle sizes of Fe and Li$_2$O after cycling exhibit no significant changes compared to those before cycling. All the above analyses confirm an electrochemically stable electrode achieved in the chemically synthesized Fe/Li$_2$O composite.

From the HRTEM characterization results, it can be inferred that the spatial distribution of the components in the composite electrode significantly influences the reaction depth and degree of completion for its subsequent conversion reaction. It is precisely due to a

confinement effect imposed by the spatial distribution of the components that the conversion is largely limited in the chemically synthesized $Fe/Li_2O$ composite. At high potentials, only a small part of Fe and $Li_2O$ undergo conversion, while a large portion of the mixed electronic/ionic conductor interface is preserved instead of being reformed during subsequent cycling at low potentials. The effective preservation of functional interfaces resulting from spatial distribution plays a crucial role in ensuring the long-term cycling stability of materials. Similar pseudocapacitive lithium storage based on spatially confined electrochemical reactions to avoid intercluster migration upon cycling has also been demonstrated in a previous study[53]. As a direct comparison, Supplementary Fig. S6a shows the cycling performance of $Fe_3O_4$ for the first 300 cycles at a current density of $1\,A\,g^{-1}$, corresponding to the electrochemical performance of the $Fe/Li_2O$ system without any specific spatial confinement. It is evident that its cycling stability is poor, with a capacity retention of only 56 % after 300 cycles at $1\,A\,g^{-1}$, much lower than the capacity retention of the spatially confined $Fe/Li_2O$ system at the same current density after 300 cycles (96%, as shown in Supplementary Fig. S6b).

To validate the effectiveness of the heterogenous interface that is responsible for most of the measured capacity, a thermodynamic analysis was conducted using the well-established "job-sharing" dual-phase heterogeneous junction charge storage model for the charge storage quantification within the low voltage range of $Fe/Li_2O$ material[53-55], as shown in Fig. 3h. Supplementary Fig. S7 shows the discharge voltage-capacity curve of $Fe/Li_2O$ in a selected low-voltage range during electrochemical measurements. Based on the fitted values of n in the vicinity of Small Q ($n \approx 3$), we could confirm the presence of an effective metal-nonmetal interface within the 0.01–1.3 V range[54,56]. Subsequently, the value of F(E,Q) was obtained according to the formula $F(E,Q) = E + n\frac{k_B T}{e}\ln Q$. By linear fitting the F(Q,E)–Q curve, we obtained a fitting value of $\gamma = 0.21$, which is in line with an effective range of magnitudes, strongly confirming the dominant role of the dual-phase heterogeneous junction space charge storage mechanism in this voltage range[53]. Furthermore, we cycled the electrode within the voltage window between 0.01 and 1.3 V in order to obtain CV curves at different scan rates (Supplementary Fig. S8a). The b values of cathodic and anodic peaks, which can be calculated from the relationship between peak current and scan rate, were 0.97 and 0.88, respectively (Supplementary Fig. S8b), further confirming that a typical surface-controlled electrochemical reaction kinetics exists within the voltage range of 0.01–1.3 V[57,58].

To obtain firm evidence that the above-mentioned charge storage mode indeed follows the space charge storage mechanism and dominates during the electrochemical process, magnetic responses of the $Fe/Li_2O$ electrode were acquired during the charging and discharging processes (Fig. 4a, b). Analyses of the temperature dependence of magnetic susceptibility ($\chi$) at high temperatures ($T > 200\,K$) (Fig. 4a) provide an important physical quantity, the Curie constant ($C$), for different voltages at various states (0.01 V, 1.3 V, and 3 V, respectively). According to the Curie–Weiss law[59]:

$$\chi = \frac{C}{T - \Theta} \qquad (2)$$

the Curie constant $C$ can be calculated by fitting the slope of the measured M–T curve. By comparing the Curie constant at different voltage values, we can clearly observe that there was no significant variation in the Curie constant from the initial state to completely discharged to 0 V and recharged to 1.3 V ($C$ remains around $1.5\,cm^3\,K\,mol^{-1}$). However, when the electrode was charged to 3 V, there was a significant increase in the value of C (to $1.82\,cm^3\,K\,mol^{-1}$). This phenomenon indicates that the intrinsic magnetic moment of the electrode has undergone a change, and the magnetic state of the relevant substance has also changed. This is consistent with the results reported in previous studies,

which affirms that the space charge storage mechanism predominates in the range of 0.01–1.3 V. Within the space charge storage voltage range, Fe maintains a stable magnetic state. However, the magnetic properties of Fe start to change from 1.3 V, indicating the occurrence of a conversion reaction above this voltage.

Figure 4b shows the M–H curves of the $Fe/Li_2O$ electrode and their corresponding Langevin fitting curves after being discharged to 0.01 V and charged back to 1.3 V at room temperature. The saturation magnetization obtained from the fitting ($87.5\,emu\,g^{-1}$ at 0.01 V and $68.0\,emu\,g^{-1}$ at 1.3 V) indicates that the change in magnetization of the electrode between the two potential points of 0.01 V and 1.3 V is $\Delta M = 21.4\,emu\,g^{-1}$. The capacity generated by space charge storage can be quantitatively estimated by the following formula[27]:

$$Q = \frac{\triangle M \times e}{3.6 \times \mathcal{P} \times \mu_B} \qquad (3)$$

In the formula, $\mathcal{P}$ represents the spin polarization value, which is obtained by $\mathcal{P} = \frac{\alpha_S \times e \times c^2}{\varepsilon \times \mu_B}$, where $\alpha_S$ denotes the surface magnetoelectric coefficient and $\varepsilon$ is the dielectric constant of the media adjacent to the Fe surface[60,61]. Using the first-principal calculation values $\alpha_S$ ($\approx 2.4 \times 10^{-14} – 2.9 \times 10^{-14}\,G\,cm^2$)[62], the surface spin polarization can be calculated as $\mathcal{P} \approx -37\% \sim -45\%$. Combining with the measured magnetization variation $\Delta M$ in the space charge storage interval, the capacity obtained through space charge storage can be estimated as: $206.00 < Q_1 < 250.48\,(mAhg^{-1})$. Based on the average capacity of $407\,mAh\,g^{-1}$ over the first 300 cycles at a low current density of $1\,A\,g^{-1}$ for the $Fe/Li_2O$ electrode (Supplementary Fig. S6b), and as all the aforementioned characterization results were obtained at this current density, the capacity provided by the space charge storage mechanism accounts for approximately 51–62% of the overall capacity.

Typically, the reaction kinetics of interfacial storage processes are much faster than those involving intercalation and phase change. Therefore, higher rates can further exploit the potential of the interface-based storage mechanism[63,64]. To highlight the interfacial storage contribution under larger currents, we extended the same set of characterization and analyses to the $Fe/Li_2O$ electrode at several higher charge/discharge rates. Supplementary Fig. S9 displays the elemental valence state changes, thermodynamic fitting analyses data, and magnetic characteristics under current densities of $5\,A\,g^{-1}$, $10\,A\,g^{-1}$, and $20\,A\,g^{-1}$, respectively. It is evident from the XPS spectra that there is still a small amount of $Fe^{2+}$ generated when charge to the high potential at a current density of $5\,A\,g^{-1}$ (Supplementary Fig. S9a), but the degree of $Fe^{2+}$ generation is significantly lower compared to that at the low current density of $1\,A\,g^{-1}$. At higher current densities like $10\,A\,g^{-1}$ and $20\,A\,g^{-1}$ (Supplementary Fig. S9b and Fig. S9c), $Fe^{2+}$ generation cannot be observed even at the fully charged potential. In other words, $Fe^0$ can maintain a higher chemical stability at high current densities. The thermodynamic analyses show that the increase of current density has no significant effect in the voltage interval where space charge storage dominates. Specifically, fitting values of $\gamma$ within the voltage range of 0.01–1.3 V at high current densities of $5\,A\,g^{-1}$, $10\,A\,g^{-1}$, and $20\,A\,g^{-1}$ are $\gamma = 0.23$, 0.28, and 0.29, respectively. These results mean that the decrease in the degree of conversion reaction under high rates does not affect interfacial charge storage, and also subtly illustrate that the space charge storage mechanism is more suitable for fast-charging working environments. By analyzing the M–H curves and fitting results of Langevin function, we calculated $\Delta M$ in the space charge storage mechanism predominated range (0.01–1.3 V) of the electrode at high current densities of $5\,A\,g^{-1}$, $10\,A\,g^{-1}$ and $20\,A\,g^{-1}$ to be 19.5, 19.1, and 15.9 ($emu\,g^{-1}$) respectively. Then the capacity realized through space charge storage is calculated as: $207.51 < Q_5 < 250.48\,(mAh\,g^{-1})$, $203.68 < Q_{10} < 247.72\,(mAh\,g^{-1})$, and $169.52 < Q_{20} < 206.17\,(mAh\,g^{-1})$. Based on the capacity values of the $Fe/Li_2O$ electrode at this current (Fig. 2b), the corresponding space charge

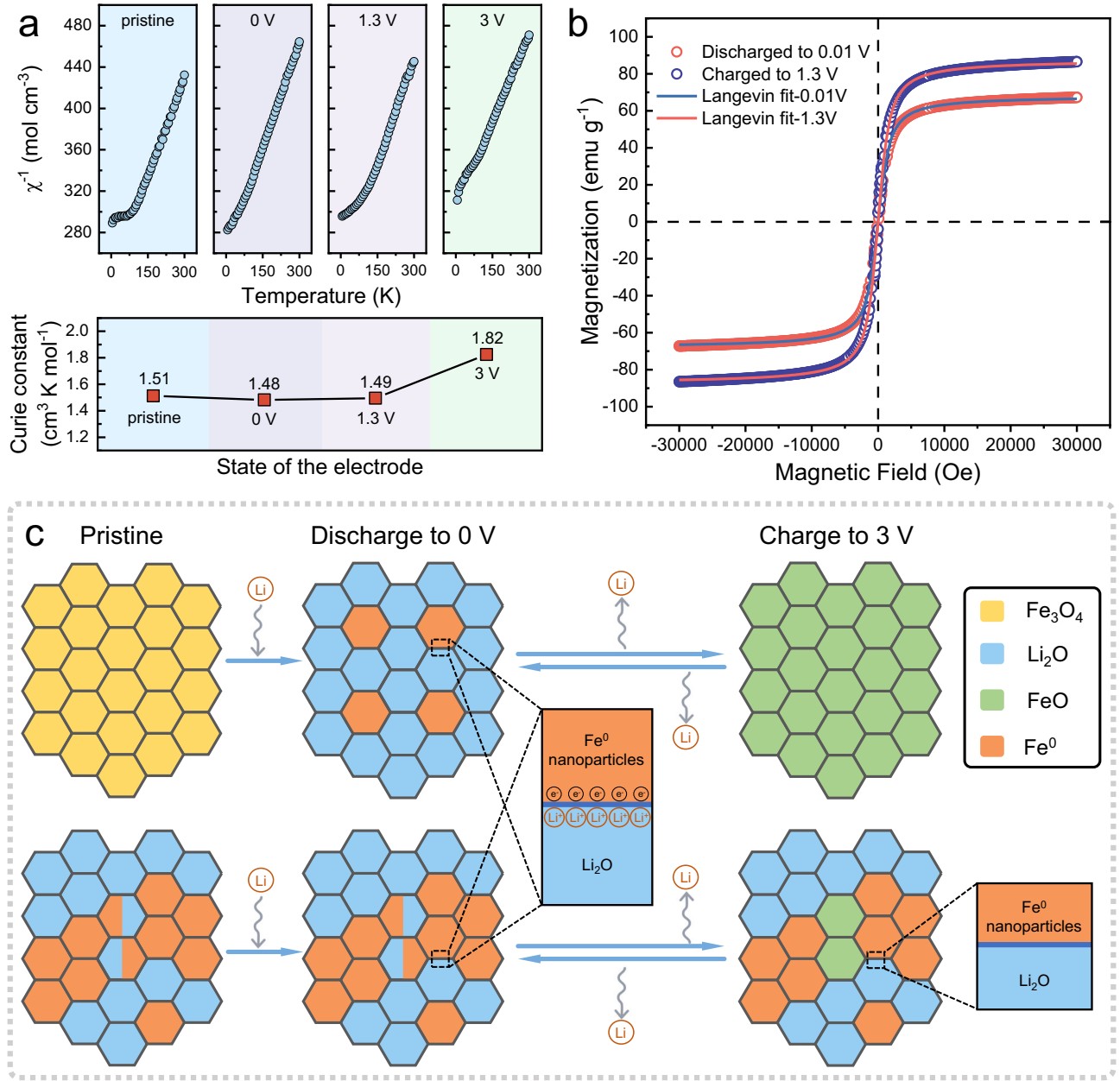

**Fig. 4 | Magnetometry characterization and reaction mechanism of the Fe/Li₂O electrode. a** M–T curves of the electrode in the initial state, discharged to 0.01 V, charged to 1.3 V, and charged to 3 V, and the corresponding Curie constant calculated by Curie–Weiss law at each state. **b** M–H curves of the electrode discharged to 0.01 V and charged to 1.3 V at room temperature and the corresponding Langevin fitting curves. **c** Schematic comparison of the lithium storage mechanism between commercial Fe₃O₄ and as-designed Fe/Li₂O.

storage capacity ratios can be estimated as 68–82%, 78–95%, and 86–104%, respectively. This means that under sufficiently large current density, the space charge storage mechanism can be responsible for nearly 100% of the capacity.

**Space charge storage mechanism and its domination dependence**

Summarizing above analyses, we propose the lithium storage mechanism in our chemically designed Fe/Li₂O as follows:

$$\mathrm{Fe/Li_2O} + x\mathrm{Li}^+ + xe^- \rightarrow \mathrm{Fe}^0 |xe^-||x\mathrm{Li}^+|\mathrm{Li_2O} \text{ (for the first full discharge)} \quad (4)$$

$$\mathrm{Fe}^0|xe^-||x\mathrm{Li}^+|\mathrm{Li_2O} \leftrightarrow \mathrm{Fe/Li_2O} + x\mathrm{Li}^+ + xe^- + \mathrm{FeO} \text{ (at low C rates)} \quad (5)$$

$$\mathrm{Fe}^0|xe^-||x\mathrm{Li}^+|\mathrm{Li_2O} \leftrightarrow \mathrm{Fe/Li_2O} + x\mathrm{Li}^+ + xe^- \text{ (at high C rates)} \quad (6)$$

In the equation, "||" represents the electronic/ionic conductor interface, and "|" represents the space charge zone in a single phase. The mechanistic schematic diagram of the entire reaction process at low C rates is shown in Fig. 4c. For the conventional iron oxide anodes such as Fe₃O₄, Fe could be fully reduced to Fe⁰ at a low potential, electrochemically forming an electronic/ionic conductor interface with the simultaneously generated Li₂O, which can provide an extra storage capacity that is physical in nature[27]. When charged to a high potential, the Fe/Li₂O generated by the electrochemical conversion reactions undergoes an oxidation reaction to form FeO. In contrast, the Fe/Li₂O material synthesized directly by chemical reduction in this work exhibits a limited conversion degree due to a special spatially

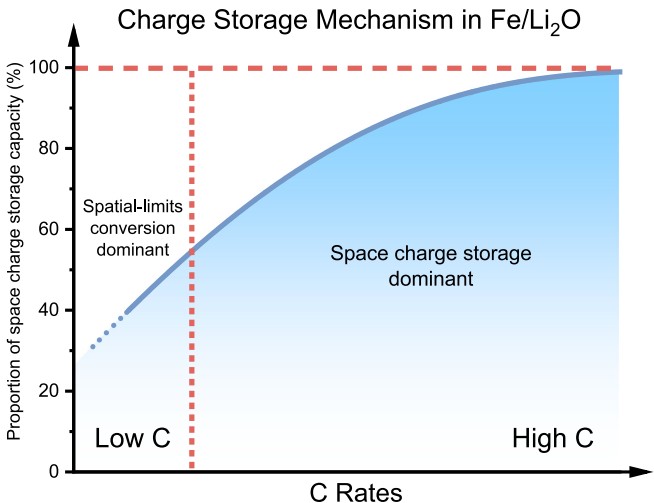

**Fig. 5 | Domination dependence of the lithium storage mechanism.** Schematic diagram showing the dependence of lithium storage mechanism on C rates in the Fe/Li$_2$O electrode.

distributed Fe and Li$_2$O. As a result, most of the Fe/Li$_2$O interfaces can be preserved during electrochemical cycling, and Fe can continuously form interfaces with the surrounding Li$_2$O or the electrolyte as an electron conductor, enabling stable space charge storage. This is exactly the underlying reason why the chemically synthesized Fe/Li$_2$O material exhibits unprecedented electrochemical performance as an anode in LIBs. As the charge–discharge rate increases, the space charge storage mechanism plays a more dominant role, eventually contributing close to 100% of the measured capacity, appearing as a full space charge storage electrode. A schematic diagram showing the rate-dependent lithium storage mechanism in the artificially constructed mixed conductor electrode is given in Fig. 5, which also demonstrates the strong relevance of the space charge storage mechanism in designing high-performance, fast-charging materials.

**The Universality**

To prove the generality of the design strategy and the storage mechanism, the similarly synthesized Fe/LiF, Fe/Li$_2$S, and Fe/Li$_3$N were characterized and evaluated similarly as the Fe/Li$_2$O and the results are given in Supplementary Fig. S10–S13. In short, the synthetic process is versatile as, in all cases, the pure phase of the targeted product was achieved (Supplementary Fig. S10). Further characterization confirms the presence of pure Fe nanoparticles in the material, consistent with the characteristic of the Fe/Li$_2$O material (Supplementary Fig. S11 and Fig. S12). According to magnetic measurements (Supplementary Fig. S13), the saturation magnetization of the Fe/LiF, Fe/Li$_2$S, and Fe/Li$_3$N is 62.9 emu g$^{-1}$, 82.3 emu g$^{-1}$, and 127.7 emu g$^{-1}$, respectively. The corresponding Fe content is calculated to be 30%, 39%, and 61%. These values are consistent with the ICP measurements within the allowable range of error (Supplementary Table S1). The particle sizes calculated from Langevin fittings are $R_{LiF} = 3.55$ nm, $R_{Li_2S} = 4.15$ nm, and $R_{Li_3N} = 3.51$ nm, which are in accordance with the values obtained from HRTEM. Based on the above characterization results, it can be concluded that a series of phase-pure iron/lithium compounds have been successfully prepared, and all of them could potentially serve as an artificial mixed electronic/ionic conductor for fast-charging/discharging anode used in LIBs.

We also conducted electrochemical measurements on electrodes fabricated from the Fe/LiF, Fe/Li$_2$S, and Fe/Li$_3$N materials. As shown in Fig. 6, they are able to deliver specific capacities of 121.6 mAh g$^{-1}$, 125.0 mAh g$^{-1}$, and 120.0 mAh g$^{-1}$ under a high current density of 50 A g$^{-1}$, respectively. Moreover, all of them can achieve stable long

cycles up to 10,000 cycles under a high current density of 10 A g$^{-1}$. The excellent electrochemical performance of this series of iron/lithium compound materials showcases the feasibility of the materials design and mechanism exploration demonstrated in this work.

In conclusion, we have designed a type of iron/lithium composite materials with high energy density, high-rate performance and high cycle stability as anodes for lithium-ion batteries. The key lies in the construction of a mixed electronic/ionic conductor in which an electrochemical stable heterogeneous interface exists and functions as decoupled and rapid charge transport pathways. Combining thermodynamic analyses and magnetic measurements, we verify that the charge storage mechanism of the electrode materials at low C rates is mainly based on space charge storage accompanied by a small amount of conversion reaction. This is confirmed through a comprehensive structural, chemical, and physical characterization, as most of the Fe$^0$ (the electronic conductor) remains unchanged during the electrochemical cycle. The otherwise detrimental conversion reaction is largely inhibited in our synthesized materials due to a confined spatial distribution of the functional components inside the composite. As the charge-discharge rate increases, the dominance of the advantageous interfacial charge storage also gradually rises, and the conversion reaction is more and more insignificant. Eventually, the electrode achieves nearly complete space charge storage mode operating only at the heterogeneous interface. This study emphasizes the critical role of interfacial effects in advancing battery development and demonstrates the potential viability of space charge storage in the future generation of fast-charging energy storage systems.

## Methods

### Synthesis of the Fe/Li$_2$O

The Fe/Li$_2$O mixed electronic/ionic conductor material was prepared using a high-temperature lithium thermal reduction method in a glove box (Mikrouna super 1220/750, Shanghai, China) filled with Ar gas to isolate it from air. Firstly, 0.18 g lithium metal foil (thickness 50 μm, 99%, Suzhou Duoduo Chemical Technology Co.) was weighed and placed in a tantalum crucible and heated on a heating stage set to 185 °C, and the lithium metal was slowly mechanically stirred until it melted. Then, 0.6 g Fe$_3$O$_4$ powder (AR, Macklin, Shanghai, China) was weighed and added into the molten lithium gradually and batch-wise, while the mechanical stirring was maintained during the addition process. After the addition was completed, the mixture was continuously stirred at 185 °C for an additional 20 min until the molten lithium was fully mixed with the Fe$_3$O$_4$ powder. Subsequently, the heating temperature was raised to 200 °C and the molten mixture underwent a solid-state calcination process, typically for a duration of 2 h. Mechanical stirring was maintained throughout the calcination process to ensure a complete reaction. After the reaction was terminated, the mixture was allowed to cool to room temperature, resulting in the formation of a black Fe/Li$_2$O solid powder. It is important to note that the final product should be avoided contacting the air or moisture and it has to be stored in the glove box.

### Synthesis of the Fe/LiF

The preparation method and experimental conditions for Fe/LiF were nearly the same as those of Fe/Li$_2$O. The difference lay in the replacement of the reactant from 0.6 g Fe$_3$O$_4$ powder to 0.13 g FeF$_3$ (AR, Sigma-Aldrich, U.S.A.), and the calcination temperature was adjusted to 240 °C. Typically, the reaction between molten lithium and FeF$_3$ could be more violent.

### Synthesis of the Fe/Li$_2$S

Again, the preparation method and experimental conditions for Fe/Li$_2$S were nearly the same as those of Fe/Li$_2$O. The reactant changed from 0.6 g Fe$_3$O$_4$ to 0.3 g FeS (AR, Macklin, Shanghai, China), and the calcination temperature was adjusted to 220 °C.

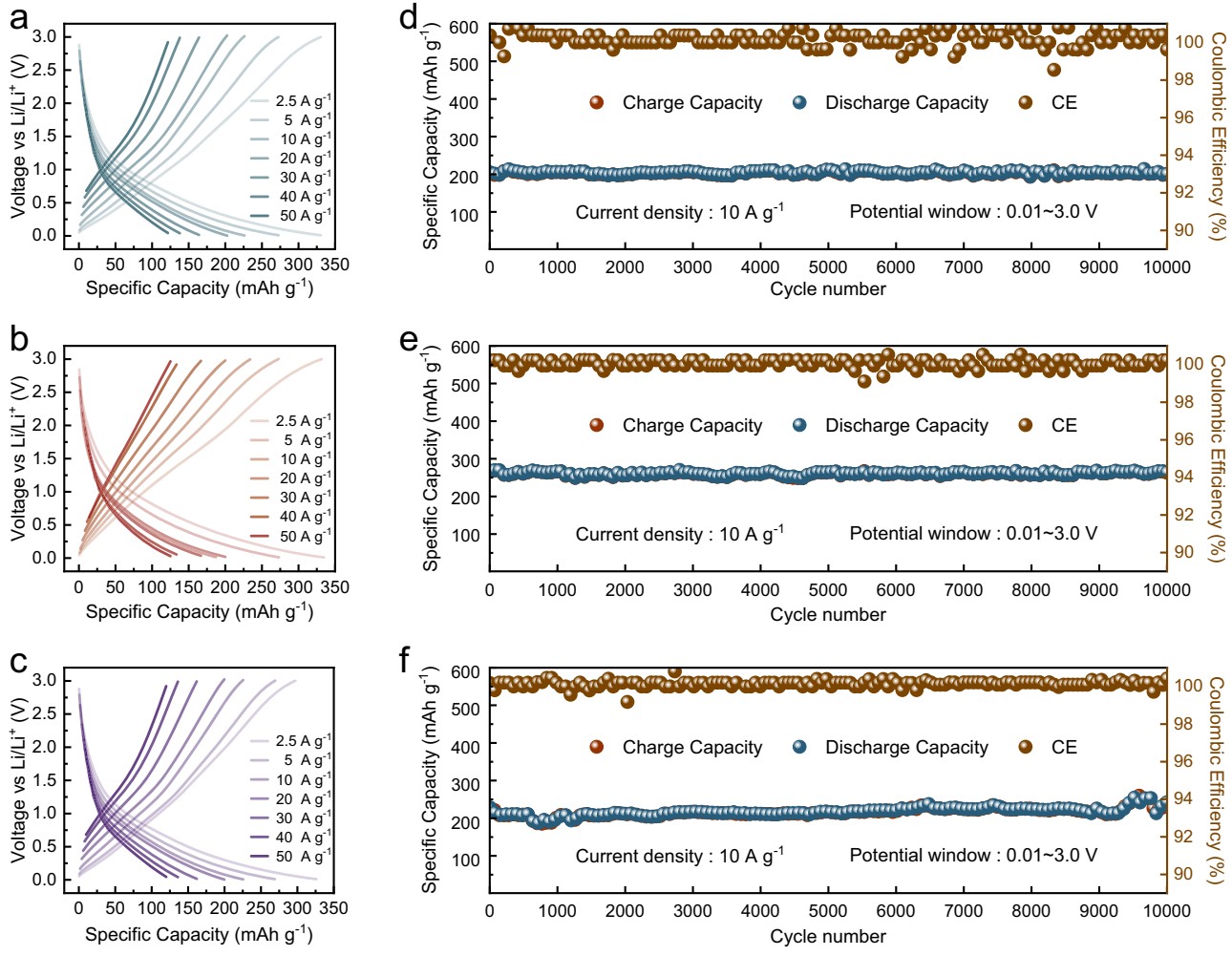

**Fig. 6 | Electrochemical performance of the Fe/Li$_x$M (M = F, S, N) electrodes. a–c** Charge−discharge curves of the Fe/LiF (**a**), Fe/Li$_2$S (**b**), and Fe/Li$_3$N (**c**) electrodes at different current densities. **d–f** Long-term cycling performance of the Fe/LiF (**d**), Fe/Li$_2$S (**e**), and Fe/Li$_3$N (**f**) electrodes at a current density of 10 A g$^{-1}$.

## Synthesis of the Fe$_2$N

Before synthesizing Fe/Li$_3$N, a precursor Fe$_2$N was prepared first: 1 g Fe$_3$O$_4$ was weighed and placed in a porcelain boat. Then the boat was subjected to NH$_3$ gas in a high-temperature tube furnace at 800 °C for 2 h. The heating and cooling rates were both set at 5 °C min$^{-1}$, then the precursor Fe$_2$N could be obtained.

## Synthesis of the Fe/Li$_3$N

After the Fe$_2$N precursor was obtained, the preparation method and experimental conditions for Fe/Li$_3$N were essentially the same as those of Fe/Li$_2$O. The reactant changed from 0.6 g Fe$_3$O$_4$ to 0.18 g Fe$_2$N and the calcination temperature was still adjusted to 200 °C.

## Materials characterization

The crystal phases of the as-synthesized products were characterized by XRD (Bruker D8 21 Advance, Germany) with Cu K$\alpha$ radiation at 40 kV and 30 mA. FESEM (ZEISS, Sigma 300) and HRTEM (JEOL, JEM-2100F) were employed to investigate the morphologies, microstructures, and element contention variations of the materials. XPS was conducted with an ESCALAB 250Xi spectrometer using an Al K$\alpha$ source to probe the surface chemical compositions and valent states of the elements of the samples. Binding energies were calibrated using carbon (C 1s = 284.6 eV). Before observation, the sample was ion-etched by Ar sputtering with an etching depth of approximately 20 nm.

## Magnetic measurements

The magnetic susceptibility measurements were performed using a Quantum Design Physical Property Measurement System (PPMS) and vibrating sample magnetometry (VSM) in the temperature range of 5–300 K under an applied field of 0.1 T. All the samples were packed into capsules to prevent exposure to the air. The obtained data were corrected considering the core diamagnetism based on Pascal's constants.

## Preparation of the working electrode

The lithium storage properties of the Fe/Li$_2$O and other iron/lithium compound materials were measured by CR2032-type coin cells with lithium metal as the counter electrodes. The electrodes were fabricated by drop-casting a slurry consisting of active materials (70 wt%), conductive carbon black (Super P, 20 wt%), and binder (polyvinylidene difluoride, PVDF, 10 wt%) onto a rounded copper current collector of 11 mm in diameter. Typical mass-loadings of ~2.5 mg cm$^{-2}$ were attainable without cracking of the electrode. A Celgard 2325 film (Whatman) was placed between the working electrodes and the lithium metal. The electrolyte of all coin cells was 1 M LiPF$_6$ in 1:1 (volume ratio) ethylene carbonate (EC) and diethyl carbonate (DEC). The volume of the electrolyte was ~90 μL. All slurry casting and cell assembly work was performed inside a glove box filled with argon gas.

## Electrochemical measurements

The galvanostatic charge/discharge (GCD) measurements of the cells were evaluated on a multichannel battery tester (NEWARE CT-4008) within the voltage range of 0.01–3.0 V. The applied current is determined based on the relevant current density (1–50 A g$^{-1}$) and the mass loading of active materials. Cyclic voltammogram (CV) tests were implemented over the range of 0.01–3.0 V on an electrochemical workstation (CHI660E). All the electrochemical measurements were carried out in an environmental chamber and the environmental temperature is 25 °C. The specific capacities were evaluated with respect to the mass of Fe in the active materials.

## Thermodynamic modeling

From the viewpoint of basic thermodynamics, in the case of interfacial charge storage at a two-phase heterojunction, the storage capacity Q over a certain voltage range depends on the lithium activity $a_{Li}$, which is a function of the cell voltage E:

$$\exp\left(-\frac{eE}{k_B T}\right) \propto a_{Li} \propto Q^n \exp(\gamma Q) \qquad (7)$$

where $e$ is the electron charge, $k_B$ is the Boltzmann constant, $T$ is the temperature, $n$ is a fitting parameter between 3 and 4, and $\gamma$ is a specific constant that contains physical properties of the junction, for example the junction length, materials density, dielectric constant, etc. According to the derivation by Fu et al.[54], a reasonable value of $\gamma$ could be estimated to be on the order of 1 g (mAh)$^{-1}$.

To verify if a charge storage process obeys the space charge storage mechanism or job-sharing mechanism within the voltage range of interest, multiple charge–discharge cycling is first conducted to reach a stable Q. Then a series of values of Q and E are obtained in one complete cycle for the subsequent fitting process. According to Eq. (7), in a small Q regime at high potentials, the factor $Q^n$ will be dominant, while for a large Q regime at low potentials, the factor $\exp(\gamma Q)$ will be dominant.

By taking the Log of both sides of Eq. (7), we have

$$\ln Q = -\frac{e}{n k_B T}E - \frac{\gamma Q}{n} - \frac{\ln k}{n} \qquad (8)$$

Therefore, the value of $n$ could be obtained by performing linear fitting of Eq. (8) in the small Q regime, and a value between 3 and 4 should be expected.

Furthermore, let $F(E,Q) = E + n\frac{k_B T}{e}\ln Q$, then:

$$F(E,Q) = -\gamma\frac{k_B T}{e}Q - \frac{k_B T}{e}\ln k \qquad (9)$$

After fitting the slope of the F(E,Q) vs. Q curve in the large Q regime, the value of $\gamma$ can be obtained. If the value of $\gamma$ is within the order of 0.1 g (mAh)$^{-1}$, it can be concluded that charge storage in the voltage range of interest at the two-phase heterojunction is indeed dominated by the space charge storage mechanism.

## Data availability

The data generated in this study are provided in the Supplementary Information/Source data file. Source data are provided in this paper.

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

## Acknowledgements

H.L. acknowledges the support from the Taishan Scholar Program of Shandong Province (tsqn202211118), the Excellent Youth Science Fund Project of Shandong China (ZR2023YQ008), Outstanding Youth Innovation Team of Universities in Shandong Province (2021KJ020), National Natural Science Foundation of China (51804173). Y.Z. acknowledges the support from the Taishan Scholar Program of Shandong Province (tsqn202306115). G.Y. acknowledges the funding support from the Welch Foundation Award F-1861 and the Camille Dreyfus Teacher-Scholar Award.

## Author contributions

H.L., Y.Z., and G.Y. conceived the project and designed the experiments. L.Z. and T.W. performed the sample preparation and characterization. L.Z. conducted electrochemical measurements. F.Z. and Y.L. performed the XPS and TEM measurements. Q.L. carried out the magnetometry measurements. L.Z. carried out the thermodynamic analyses. H.L., L.Z., and Y.Z. analyzed the data. L.Z., Y.Z., Z.J., H.L., and G.Y. co-wrote the paper, and all authors contributed to it.

## Competing interests

The authors declare no competing interests.
