## [Peer Review File · Nature Communications]

A fast-charging/discharging and long-term stable artificial electrode enabled by space charge storage mechanismREVIEWER COMMENTS

Reviewer #1 (Remarks to the Author):

The authors designed an iron/lithium composite anode for lithium-ion batteries. Due to the construction of a mixed electronic/ionic conductor, interfacial charge storage mechanism occurs at the high rate, which is confirmed by the thermodynamic analyses and magnetic measurements. Overall, this work provides insights into the rapid-charge electrochemical energy storage systems. The manuscript is well organized and written. Several issues need to be addressed before publication, which are detailed below.

My first and foremost concern is the electrode loading (mg/cm^2 ; mAh/cm^2). If it is too low ($< 2 \text{ mAh}/\text{cm}^2$), its importance for the practical application will be largely reduced. Please provide that and discuss more.

Besides, similar charge storage mechanism has been reported previously (J. Electrochem. Soc. 2004, 151, A1878; Phys. Chem. Chem. Phys. 2009, 11, 9497). The authors are suggested to indicate clearly the novelty. The performances of these materials should be compared in Figure 2e.

Finally, the authors adopted HRTEM to demonstrate the existence form of Fe/Li₂O. While HRTEM is a micro-region analysis technique, can the authors provide the method to obtain the bulk features? More evidences are required to demonstrate the mechanism of Figure 1b.

Reviewer #2 (Remarks to the Author):

The authors have provided a super amazing C-rate (100 C and 500C etc.) for fast-charging and potentially high power density Li-based cell designs based on a MIEC concept. Overall, I believe the performance achieved is among the top of the state-of-art results in the battery field. Certainly it deserves the quality for a publication in Nature Communications after carefully resolving my comments below (minor revision is needed):

1. The authors need to mention the mass loading of the active materials used for all the redox chemistry and the overall cathode loading and should highlight in the abstract. As the authors main focus on the manuscript is the fast charging/discharging capability, it need to do a direct comparison about the power density (cathode level and cell level) achieved in this manuscript and the published results in the similar fields. The authors also encouraged to mention N/P ratio and how much liquid electrolyte was used in building the cells. In addition, as different manuscript calculated power density in different ways, the authors need to state the detailed calculation steps/approaches in the supporting information.
2. As the authors measured the electrodes are MIEC, have the authors measured the ionic and electronic conductivity of the cathode active materials themselves?
3. What are the rationals for choosing $M = \text{O}, \text{F}, \text{S}, \text{N}$?
4. When the authors described in the introduction: "Thermodynamic fitting and magnetic analysis, combined with structural characterization, prove the presence of only limited conversion reaction due to reactant spatial distribution confinement even at low rates", how low the C-rate threshold is?

5. Although LiF is a very ionic and electronic insulating materials, the authors have achieved superior cycling performance (Figure 6a and 6d). The authors need to comment on how the superior cycling was achieved based on an insulating nature of LiF. Was the Li storage mechanism in Fe/LiF the same as Fe/Li₂O cathode? Furthermore, as the authors use MIEC concept and low mass loading, the authors need to explain why high carbon content (20%) was used.

6. The authors need to explain in detail how MIEC was incorporated in the Fe/Li₂M materials, illustrating the ionic conduction and electronic conduction contributors in the electrode.

Reviewer #3 (Remarks to the Author):

In this work, ultra-fast charging/discharging, ultra-stable and high energy charge storage properties can be realized in an artificial electrode made from a mixed electronic/ionic 21 conductor material (Fe/Li_xM, where M=O, F, S, N) enabled by a space charge principle. As a result, the Fe/Li₂O electrode is able to be charged/discharged to 126 mAh g⁻¹ in 6 s at a high rate of up to 50 A g⁻¹. The authors claimed that the space charge storage mechanism will play a critical role in advancing electrochemical energy storage and provides a unique perspective for designing high-performance anode materials for lithium-ion batteries. However, some critical problems are existed in the manuscript.

(1) In 2017, Yingzhu Jiang group designed SnO₂-Fe₂O₃-Li₂O nanocomposite (Advanced Materials, 2017, 29, 1606499), and the high fraction of interfaces of Fe/Sn/Li₂O during the charge/discharge processes ensure a reversible capacity of 350 mAh g⁻¹ can be maintained at an ultrahigh current density of 80 A g⁻¹, much higher than this work (126 mAh g⁻¹ at 50 A g⁻¹). In Jiang's work, the combination of pseudocapacitive lithium storage and spatially confined electrochemical reactions were attributed to the excellent rate performances for the Sn-based nanocomposite anode materials. However, only space charge storage mechanism is demonstrated for the enhanced rate performances, which may be not enough.

(2) The description of Figure 1a-b in line 81-82 is too simple and ambiguous, and more explanations are needed.

(3) In the charge state of 3.0 V, no characteristic peaks of FeO can be observed, which may be attributed to the low crystallinity of the charge products. However, obvious peaks of FeO can be observed in the XPS in the Figure 3a, and the ratio of FeO should be calculated. Furthermore, the remarkable formation of FeO is observed at high current density of 10 C. Undoubtedly, the ratio of FeO will be greatly increased if the current was set at 0.1 C to the charge state of 3.0 V, which can be characterized by the XPS, and the limited conversion reaction may not correct. Why did the author choose 10 C as the charge current, in this condition? Many important structural and phase change information will miss due to the slow interfacial dynamics and large overpotential under high rates.

(4) According to proposed the lithium storage mechanism, FeO was the undesirable product in the space charge storage mechanism, and the readers may wonder that whether the FeO will accumulate, and therefore the amount of FeO after intensive cycles should be distinguished.

(5) In the Figure 6a and Figure 2c, obvious pseudocapacitive lithium storage is observed, which ensure the excellent rate performances. The particle size of the Fe/Li₂O before cycling and after cycling should be added.

(6) The cycle performance and rate performance were obtained at high current densities, and what about the electrochemical performances of the Fe/Li₂O at small current density

such as 1 C? Besides, what about the loading amount of Fe/Li₂O? Did the authors evaluate the electrochemical performances in the full cell, which is very important for its practical applications. Furthermore, the comparison in Figure 2f is not fair because of different loading amount.

(7) As shown in Figure 3 and Figure 4, the authors provided a large number of data to prove the role of mixed electronic/ionic conductor interface in stabilizing the electrochemical performances, however, there is no any data about the role of spatial distribution in stabilizing the electrochemical performances.

(8) What about the initial coulombic efficiency at low current density? What about the large-scale production of the Fe/Li₂O anode?

Point-to-point Response to Reviewers' Comments

No. NCOMMS-23-62207-T

Title: An ultrafast-charging/discharging and stable artificial electrode enabled by space charge storage mechanism

First of all, we want to thank all the three reviewers for their positive evaluation of our manuscript with their detailed and insightful suggestions. We appreciate the acknowledgements from the reviewers that this work “provides insights into the rapid-charge electrochemical energy storage systems” and “deserves the quality for a publication in Nature Communications”. As presented below, all the points raised by the reviewers have been carefully addressed with an extensive set of newly added experiments and analyses.

Response to Reviewer #1:

Reviewer #1: The authors designed an iron/lithium composite anode for lithium-ion batteries. Due to the construction of a mixed electronic/ionic conductor, interfacial charge storage mechanism occurs at the high rate, which is confirmed by the thermodynamic analyses and magnetic measurements. Overall, this work provides insights into the rapid-charge electrochemical energy storage systems. The manuscript is well organized and written. Several minor issues need to be addressed before publication, which are detailed below.

Response: Thank you very much for recognizing the insights provided by this work and speaking highly of our manuscript. We have addressed them with the following detailed responses.

Question 1. My first question is the electrode loading (mg/cm^2 ; mAh/cm^2). Please provide that and discuss more.

Response: We thank the reviewer for raising this question, which was also raised by the other reviewers. We want to first mention that the electrode preparation in this work was done by the so-called drop-casting method in the glove box right after synthesis of the active materials, because they are air sensitive. This method is facile but is not very effective in producing thick electrodes (due to cracking problem) compared to the doctor-blade process, which we unfortunately are not able to perform in the glove box at current stage. Based on the drop-casting method, the representative electrode loading is usually within the range of $\sim 2.5 \text{ mg cm}^{-2}$, which corresponds to an areal capacity of $\sim 1.0 \text{ mAh cm}^{-2}$ at a current density of 10 C, and $\sim 0.65 \text{ mAh cm}^{-2}$ at a current density of 100 C. Although currently limited by the preparation method, our mass loadings of the active material are still on par with the current state-of-the-art anode materials for fast charging lithium-ion batteries (as shown in Table R1) and fall well within the typical lab-level range. We agree with the reviewer that low electrode loading may hamper the practical application and in our case further process optimization will be needed to improve the loading. Nevertheless, beyond the engineering optimization, the scientific insights demonstrated in this work are sufficient to justify its potential demonstrated by this class of materials based on novel spin capacitance mechanisms for designing fast-charging batteries.

Table R1. The electrode loadings of some state-of-the-art anode materials for fast charging lithium-ion batteries.

Anode material	Mass loading	Reference
Li ₃ V ₂ O ₅	2~3 mg cm ⁻²	Nature 2020 , 585, 63–67.
m-SiO _x	2.45 mg cm ⁻²	Nat. Energy 2023 , 8, 129–137.
HNMG (high quality nitrogen-doped mesoporous graphene)	1 mg cm ⁻²	Nat. Commun. 2019 , 10, 1474.
3D-Ge/C	2 mg cm ⁻²	Energy Environ. Sci. 2015 , 8, 3577–3588.
pd-TNO (titanium niobate)	2.0 mg cm ⁻²	Adv. Energy Mater. 2022 , 12, 2201130.
Sr ₂ V ₂ O ₇ ·H ₂ O	1.6~1.8 mg cm ⁻²	Adv. Funct. Mater. 2023 , 33, 2214667.
KS6 graphite	2.5 mg cm ⁻²	Nano-Micro Lett. 2023 , 15, 215.

Changes made:

On page 2 in the Abstract, the following sentence has been revised: “Particularly, the Fe/Li₂O electrode is able to be charged/discharged to 126 mAh g⁻¹ in 6 s at a high rate of up to 500 C (50 A g⁻¹), and it also shows ultra-stable cycling performance for 30,000 cycles at a cycling rate of 100 C, with a mass-loading of ~2.5 mg cm⁻² of the electrode materials.”

On page 29 in the Methods section, the following sentences have been revised and added: “The electrodes were fabricated by drop-casting a slurry consisting of active materials (70 wt %), conductive carbon black (Super P, 20 wt %), and binder (polyvinylidene difluoride, PVDF, 10 wt %) onto a rounded copper current collector of 11 mm in diameter. Typical mass-loadings of ~2.5 mg cm⁻² were attainable without cracking of the electrode.”

Question 2. Some similar charge storage mechanism has been reported previously (J. Electrochem. Soc. 2004, 151, A1878; Phys. Chem. Chem. Phys. 2009, 11, 9497). The authors are suggested to indicate the novelty. The performances of these materials should be compared in Figure 2e.

Response: We sincerely appreciate this valuable comment, as it helps us extend our literature review to include earlier efforts under similar topic and also highlight the novelty of this work. Because our work is theoretically based on J. Maier’s interfacial storage model, it is actually not surprising to find earlier studies exploring similar storage mechanism from his group. Indeed, the utilization of decomposition products of metal fluorides/oxides and the direct utilization of LiF/Ti in these two references share similar space charge storage mechanism as in our LiM/Fe. However, these previous works mainly focused on electrochemical characterization. Building on earlier studies, our work innovatively employed a straightforward lithium thermal displacement reaction (as different to the previous methods using electrochemical in-situ generation or pulse laser deposition) to construct a mixed electronic/ionic conductor electrode material and realized ultrafast charging/discharging rate

and ultra-stable cycling ability. More importantly, we also successfully established a clear relationship between the contribution ratio of the space charge storage mechanism and the charge/discharge C-rate, revealing the influence of spatial confinement on the extent of detrimental conversion reactions. Based on the performance results, the mixed electronic/ionic conducting interface system constructed through the lithium thermal displacement reaction is significantly different from the electrochemically in-situ generated mixed conductor systems or physically deposited thin film systems. Additionally, beyond electrochemical characterization, we provided solid experimental evidence, including quantitative description of the storage degree through a unique magnetic probe, to prove the existence of space charge storage mechanism.

As suggested by the reviewer, the materials performance from the two references has been selectively added and compared in Fig. 2e. We thank the reviewer again for this suggestion.

Changes made:

On page 9 in the revised main text, Fig. 2e is now updated as shown below.

Fig. R1. Overall performance comparison of the Fe/Li₂O electrode with relevant reports of fast charging materials in CR (%), SCH (mAh g⁻¹), SCL (mAh g⁻¹), Rat. (C), Cyc. (number), FC (mAh g⁻¹).

On page 8 in the revised main text, the corresponding references have been added: “Such excellent electrochemical performance places the current Fe/Li₂O favorably among many other fast-charging/discharging materials, including earlier studies employing the similar storage mechanism (Fig. 2e) [41–47].”

[46] H. Li, P. Balaya, J. Maier, *J. Electrochem. Soc.* **2004**, *151*, A1878.

[47] X. Q. Yu, J. P. Sun, K. Tang, H. Li, X. J. Huang, L. Dupont, J. Maier, *Phys. Chem. Chem. Phys.* **2009**, *11*, 9497.

On page 7 in revised supplementary information, Table S3 with the corresponding references has been added as shown below.

Table R2 | Detailed information of the relevant reports of fast charging materials in terms of capacity retention (CR), specific capacity at high rate (SCH), specific capacity at low rate (SCL), rate ability (Rat.), cycle ability (Cyc.) and final capacity (FC).

Materials	CR (%)	SCH (mAh g ⁻¹)	SCL (mAh g ⁻¹)	Rat. (C)	Cyc. (number)	FC (mAh g ⁻¹)	Reference
lithium yttrium titanate (LYTO)	98	87	210	100	3000	180	[5]
Rock salt L ₃ V ₂ O ₅	87	109	266	200	1000	133	[6]
TiO ₂ /3D Cu-GM	88	94	284	60	1000	83	[7]
NiNb ₂ O ₆	78	50	244	100	20000	47	[8]
Li ₄ Ti ₅ O ₁₂ (LTO-600)	88	146	181	100	1500	150	[9]
TiF ₃ (Ti/LiF)	59	44	64	6	15	240	[10,11]

[10] H. Li, P. Balaya, J. Maier, *J. Electrochem. Soc.* **2004**, *151*, A1878.

[11] X. Q. Yu, J. P. Sun, K. Tang, H. Li, X. J. Huang, L. Dupont, J. Maier, *Phys. Chem. Chem. Phys.* **2009**, *11*, 9497.

Question 3. Finally, the authors adopted HRTEM to demonstrate the existence form of Fe/Li₂O. While HRTEM is a micro-region analysis technique, can the authors provide the method to obtain the bulk features? More discussions/evidence are needed to demonstrate the mechanism of Figure 1b.

Response: We thank the reviewer for pointing this out, which enables us to clarify the roles of various characterization techniques used in the manuscript. First, the successful formation of Fe/Li₂O via the lithium thermal displacement reaction, or the existence of Fe/Li₂O is unambiguously shown by XRD (Fig. 1e). While SEM gives the general morphology of the mixture in bulk (Fig. 1e inset), HRTEM shows clearly the intimate interface between Fe and Li₂O at a local level (Fig. 3e). The latter already partially implies the working mechanism depicted in Fig. 1b. To provide the bulk features as mentioned by the reviewer, our magnetic characterization technique, including the measurement of M-H and M-T curves, the relevant Langevin fitting (Fig. 1f and 4b) and the determination of the Curie constant (Fig. 4a), are all aimed to serve this purpose by probing directly on the bulk materials and extracting macroscopic properties. The other bulk features we could think of, which come from the performance per se, is the electrochemical properties of the Fe/Li₂O. Therefore, thermodynamic analyses and establishment of the job-sharing model were conducted on the macroscopic electrochemical system, reflecting the bulk features of the materials. Based on our previous research (*Nat. Mater.* **2021**, *20*, 76–83) and fundamental studies by other researchers (*Phys. Rev. Lett.* **2014**, *112*, 208301 and *Nature* **2016**, *536*, 159–164), conclusions drawn from the magnetic and electrochemical characterization as well as the perfectly matching thermodynamic model analysis provide strong support to the proposed mechanism depicted in Fig. 1b, namely, the obvious existence of the space charge storage mechanism in a bulk mixed electronic/ionic conducting material.

It should be pointed out that bulk features of this type of materials can be also characterized in thin film configurations such as in *Phys. Chem. Chem. Phys.* **2009**, 11, 9497. However, our materials configuration exhibits much higher performance due to the much larger portion of the functional interface inside the materials. Finally, please refer to our response to **Question 2** from **Reviewer #2**, and the measured conductivities could also serve as a bulk property of the material.

Response to Reviewer #2:

Reviewer #2: The authors have provided a super amazing C-rate (100 C and 500C etc.) for fast-charging and potentially high power density Li-based cell designs based on a MIEC concept. Overall, I believe the performance achieved is among the top of the state-of-art results in the battery field. Certainly it deserves the quality for a publication in Nature Communications after carefully resolving my comments below (minor revision is needed):

Response: Thank you very much for your favorable review of our work and your recommendation for its publication. We have addressed your comments with following detailed point-to-point response and revised our manuscript accordingly.

Question 1: The authors can provide the mass loading of the active materials and the overall cathode loading and could highlight in the abstract. As the authors main focus on the manuscript is the fast charging/discharging capability, it is helpful to do a direct comparison about the power density (cathode level and cell level) achieved in this manuscript and the published results in the similar fields. The authors also encouraged to mention N/P ratio and how much liquid electrolyte was used in building the cells. In addition, the authors need to state the detailed calculation steps/approaches in the supporting information.

Response: We sincerely appreciate the reviewer's thoughtful comments and constructive suggestions concerning the performance evaluation, as it is of high importance for the battery community. To compare the power density at cell level and also to answer partially **Question 6** from **Reviewer #3**, we fabricated a full cell using our Fe/Li₂O as anode and LiFePO₄ as cathode. As answered in **Question 1** from **Reviewer #1**, the mass loadings of the active material are usually ~2.5 mg cm⁻² (evaluated based on the mass of Fe), while the cathode material (LiFePO₄) loading is ~5.5 mg cm⁻² in the full cell in order to balance the capacities. The details regarding the mass loading of the active materials and the full cell have been supplemented in the abstract, method and supplementary information.

For the calculation of power density of the cathode in half-cell batteries, we adopted the average power method (calculating the energy released per unit time) to determine the energy density and average power density of the material (*J. Mater. Chem. A* **2016**, 4, 8716–8723; *Electrochim. Acta* **2016**, 196, 603–610 and *J. Mater. Chem. A* **2013**, 1, 6145). This decision is based on the precision of the testing equipment and sampling frequency, as well as to facilitate comparison with other materials. The detailed calculation method for the power density and the comparison with other materials published in the fast-charging field are now included in the supplementary information.

As stated above, we assembled a Fe/Li₂O||LiFePO₄ full cell to characterize the power density at the cell level. The power density of the full cell was also calculated using the average power method. The discussion, relevant data and detailed information (including the N/P ratio and the amount of liquid electrolyte used) related to the construction of the full cell and the comparative data on power density have been included in the main text and the supplementary information (Supplementary Note 2 and Supplementary Fig. S2).

Changes made:

On page 8 in the revised main text, the following sentences have been added: “In addition, to evaluate the practical capability of the designed Fe/Li₂O anode, a coin-type Fe/Li₂O||LiFePO₄ full cell was

fabricated and examined. For detailed fabrication and performance of the full cell, please refer to the Supplementary Note 2 and Fig. S2. Furthermore, we compared the energy density and power density of the Fe/Li₂O electrode and the Fe/Li₂O||LiFePO₄ full cell with representative previously published results on fast-charging materials and devices. Both the electrode and the full cell based on the Fe/Li₂O material exhibit clear power density advantages. The detailed data, including the calculation method for power density, are given in Supplementary Note 3 and Fig. S3.”

On page 29 in the Methods section, the following sentence has been added: “The volume of the electrolyte was ~90 μ L.”

On page 2-4 in revised SI, the following Supplementary Notes have been added:

Supplementary Note 2: Fabrication and performance evaluation of the Fe/Li₂O||LiFePO₄ full cell.

The Fe/Li₂O was paired with a LiFePO₄ cathode to make a full cell. The cathode electrode slurry was prepared by mixing LiFePO₄ (AR, Macklin, Shanghai, China), Super-P and PVDF binder dissolved in N-methyl-2-pyrrolidone (NMP) with the weight ratio of 80:10:10. The slurry was pasted onto an aluminum foil and the cathode was cut into 11-mm diameter discs with LiFePO₄ active material loading of ~5.5 mg cm⁻². A Celgard 2325 film (Whatman) and the 1 M LiPF₆ in 1:1 (volume ratio) ethylene carbonate (EC) and diethyl carbonate (DEC) was used as separator and electrolyte, respectively. The volume of the electrolyte was ~90 μ L. The Fe/Li₂O were firstly electrochemical prelithiation in a half-cell configuration (vs metal Li) before used as anode in full cells. The CR2032-type full cells were assembled in a glove box filled with argon gas with the LiFePO₄ electrode as cathode and the prelithiation Fe/Li₂O electrode as anode. Galvanostatic charge/discharge measurements were performed between 1.0 to 3.0 V at the current rate of 1 C (1 C = 100 mA g⁻¹). Specific capacities were calculated based on the cathode materials of each electrode.

The negative-to-positive (N/P) electrode capacity ratio was around 1.1 for the Fe/Li₂O||LiFePO₄ full cell. The initial charge voltage curve of the Li||LiFePO₄ half-cell under 1 C and the discharge voltage curve of the Li||LiFePO₄ half-cell under 1 C were used to simulate the charge voltage profile of a Fe/Li₂O||LiFePO₄ full cell. LiFePO₄ has an initial charge specific capacity of 0.94 mAh while Fe/Li₂O delivers 1.03 mAh upon discharge to 0.1 V versus Li/Li⁺. The voltage range of Fe/Li₂O is 0.1-3.0 V and its capacity is normalized to 1.1 times the LiFePO₄ capacity. The mass loading of the cathode and anode were controlled to realize a negative-to-positive electrode capacity ratio of 1.1 (Supplementary Fig. S2a). The rate capability of the Fe/Li₂O||LiFePO₄ full cell is shown in Supplementary Fig. S2b, the reversible specific capacities of 166, 127, 110, 100 and 93 mAh g⁻¹ at 1, 2, 3, 4 and 5 C rates are obtained, respectively. The capacity of the Fe/Li₂O||LiFePO₄ full cell after the 500th cycle is 140 mAh g⁻¹, resulting in a capacity retention of 85 % and an average Coulombic efficiency of >96 % (Supplementary Fig. S2c and S2d).

Supplementary Note 3: Calculation method for power density.

To compare Fe/Li₂O electrodes with previously reported high-rate anode materials, a Ragone plot is shown in Supplementary Fig. S3. Its specific energy density (E , Wh kg⁻¹) and power density (P , kW kg⁻¹) are calculated from the equations as follows:

$$P = \frac{Q\Delta E}{m\Delta t} = \frac{i\Delta E}{m} \quad (1)$$

$$E = P\Delta t \quad (2)$$

Where $\Delta E = \frac{E_{max} + E_{min}}{2}$. In these equations, Q (mAh), i (A), m (g), and Δt (s) denote the charge delivered during discharge, discharge current, mass of active materials, and discharge time, respectively. E_{max} and E_{min} are the initial and final potentials of discharge curves of galvanostatic cycling at different current densities^[1-3]. It can be seen that Fe/Li₂O exhibits higher energy densities for the full power density range outperforming all other reported materials (Supplementary Fig. S3a). Even at a power density as high as 113.4 kW kg⁻¹, it still retains an energy density of 189 Wh kg⁻¹. Supplementary Fig. S3b represents the Ragone plot of the Fe/Li₂O||LiFePO₄ full cell and other previously reported high-rate hybrid devices. The specific energy density (E , Wh kg⁻¹) and power density (P , kW kg⁻¹) are calculated by the aforementioned equations (1) and (2). It can be seen that the full cell achieves a maximal high-level energy density of 194.4 Wh kg⁻¹ at 8.4 kW kg⁻¹. The full cell shows energy preponderance when compared with many excellent reported devices.

On page 10-11 in revised SI, Supplementary Fig. S2 and S3 have been added as shown below.

Fig. R2. The electrochemical performance of Fe/Li₂O||LiFePO₄ full cell. (a) The initial charge voltage curve of a Li||LiFePO₄ half-cell under 1 C and the discharge voltage curve of a Li||Fe/Li₂O half-cell under 1 C were used to assemble Fe/Li₂O||LiFePO₄ full cell. The negative-to-positive electrode capacity ratio was around 1.1. (b) Rate capability of Fe/Li₂O||LiFePO₄ full cell at different rates (from 1 to 5 C). (c) Voltage profiles over the course of 500 cycles for a voltage window of 1.0–3.2 V. (d) The cycling stability of the Fe/Li₂O||LiFePO₄ full cell at 1 C for 500 cycles.

Fig. R3. Ragone plot comparing Fe/Li₂O electrode (a) and Fe/Li₂O||LiFePO₄ full cell (b) to published results in the similar fields^[12–23].

On page 22-23 in revised supplementary information, the following new references have been added.

- [2] L. Que, Z. Wang, F. Yu, D. Gu, *J. Mater. Chem. A* **2016**, *4*, 8716–8723.
- [3] V. Aravindan, N. Shubha, W. C. Ling, S. Madhavi, *J. Mater. Chem. A* **2013**, *1*, 6145–6151.
- [4] X. Han, P. Han, J. Yao, S. Zhang, X. Cao, J. Xiong, J. Zhang, G. Cui, *Electrochim. Acta* **2016**, *196*, 603–610.
- [12] K. J. Griffith, K. M. Wiaderek, G. Cibin, L. E. Marbella, C. P. Grey, *Nature* **2018**, *559*, 556–563.
- [13] X. Jin, Y. Han, Z. Zhang, Y. Chen, J. Li, T. Yang, X. Wang, W. Li, X. Han, Z. Wang, X. Liu, H. Jiao, X. Ke, M. Sui, R. Cao, G. Zhang, Y. Tang, P. Yan, S. Jiao, *Adv. Mater.* **2022**, *34*, 2109356.
- [14] L. Shen, S. Chen, J. Maier, Y. Yu, *Adv. Mater.* **2017**, *29*, 1701571.
- [15] J. Yan, A. Sumboja, E. Khoo, P. S. Lee, *Adv. Mater.* **2011**, *23*, 746–750.
- [16] W. Wu, M. Liu, Y. Pei, W. Li, W. Lin, Q. Huang, M. Wang, H. Yang, L. Deng, L. Yao, Z. Zheng, *Adv. Energy Mater.* **2022**, *12*, 2201130.
- [17] S. Deng, H. Zhu, B. Liu, L. Yang, X. Wang, S. Shen, Y. Zhang, J. Wang, C. Ai, Y. Ren, Q. Liu, S. Lin, Y. Lu, G. Pan, J. Wu, X. Xia, J. Tu, *Adv. Funct. Mater.* **2020**, *30*, 2002665.
- [18] Z. Yao, X. Xia, Y. Zhang, D. Xie, C. Ai, S. Lin, Y. Wang, S. Deng, S. Shen, X. Wang, Y. Yu, J. Tu, *Nano Energy* **2018**, *54*, 304–312.
- [19] A. Varzi, D. Bresser, J. Von Zamory, F. Müller, S. Passerini, *Adv. Energy Mater.* **2014**, *4*, 1400054.
- [20] X. Fu, H. Duan, L. Zhang, Y. Hu, Y. Deng, *Adv. Funct. Mater.* **2023**, *33*, 2308022.
- [21] B. Wang, W. Li, T. Wu, J. Guo, Z. Wen, *Energy Storage Mater.* **2018**, *15*, 139–147.
- [22] P. Xiong, L. Peng, D. Chen, Y. Zhao, X. Wang, G. Yu, *Nano Energy* **2015**, *12*, 816–823.
- [23] Z. Wang, M. Yao, H. Luo, C. Xu, H. Tian, Q. Wang, H. Wu, Q. Zhang, Y. Wu, *Small* **2024**, *20*, 2306428.

Question 2: As the authors measured the electrodes are MIEC, have the authors measured the ionic and electronic conductivity of the cathode active materials themselves?

Response: We thank the reviewer for pointing this loophole out, as indeed we claimed the electrodes to be MIEC mainly based on the structural characterization without checking their conductivities. Taking Fe/Li₂O as an example, we proceeded to measure the electronic and ionic conductivity using cold-pressing pellets and following the well-established AC and DC methods (*Ionics*, **2002**, *8*, 300–

313). Again, due to its air sensitive nature, preparation of the sample and all subsequent measurements were conducted under an inert gas atmosphere. A VICTOR 4090A LCR digital bridge was used for these tests.

Theoretically, the measured impedance in AC mode (frequency of 100-1000 Hz) would be the combined impedance of electronic and ionic conduction. Whereas in DC mode, the contribution of ionic conductivity would rapidly decrease (we utilized a Cu electrode, which acts as a blocking electrode to the Li^+ in DC mode). Thus, the stable results in DC mode would only reflect the contribution from electronic conductivity. As the measurement results in AC mode represent the mixed electronic and ionic conductivity, the ionic conductivity can be calculated by subtracting the electronic conductivity obtained in DC mode.

It should be noted that the preparation of the test specimen by cold-pressing may result in loosely-bound grains and possibly numerous voids, which could cause instability in the test results. To minimize such errors, we conducted multiple measurements and obtained the average values. The final test results are presented in Table R3. The calculated average electronic conductivity of the Fe/Li₂O material is $3.43 \times 10^{-6} \text{ S m}^{-1}$, while its ionic conductivity is $1.62 \times 10^{-5} \text{ S m}^{-1}$ (the total electrical conductivity of the Fe/Li₂O material is $1.96 \times 10^{-5} \text{ S m}^{-1}$). Again, due to the restriction of the cold-pressing method and the extremely small size of the constituting components in the material, these values could only serve as a rough estimation (lower limit) of the conductivities.

Changes made:

On page 6 in the revised main text, the following sentences have been added: “Considering that the material can be regarded as a mixed ionic-electronic conductor at the macroscopic scale, we have also characterized the electronic and ionic conductivities of the as-prepared Fe/Li₂O. For detailed information, please refer to Supplementary Note1 and Table. S2.”

On page 1 in revised supplementary information, the following Supplementary Note has been added:
Supplementary Note 1: The ionic and electronic conductivities of the Fe/Li₂O.

We measured the electronic and ionic conductivity using cold-pressing pellets and following the well-established AC and DC methods^[1]. Due to its air sensitive nature, preparation of the sample and all subsequent measurements were conducted under an inert gas atmosphere. A VICTOR 4090A LCR digital bridge was used for these tests.

Theoretically, the measured impedance in AC mode (frequency of 100-1000 Hz) would be the combined impedance of electronic and ionic conduction. Whereas in DC mode, the contribution of ionic conductivity would rapidly decrease (we utilized a Cu electrode, which acts as a blocking electrode to the Li^+ in DC mode). Thus, the stable results in DC mode would only reflect the contribution from electronic conductivity. As the measurement results in AC mode represent the mixed electronic and ionic conductivity, the ionic conductivity can be calculated by subtracting the electronic conductivity obtained in DC mode.

It should be noted that the preparation of the test specimen by cold-pressing may result in loosely-bound grains and possibly numerous voids, which could cause instability in the test results. To minimize such errors, we conducted multiple measurements and obtained the average values. The final test results are presented in Supplementary Table S2. The calculated average electronic conductivity of the Fe/Li₂O material is $3.43 \times 10^{-6} \text{ S m}^{-1}$, while its ionic conductivity is $1.62 \times 10^{-5} \text{ S m}^{-1}$ (the total electrical conductivity of the Fe/Li₂O material is $1.96 \times 10^{-5} \text{ S m}^{-1}$). Again, due to the restriction of the

cold-pressing method and the extremely small size of the constituting components in the material, these values could only serve as a rough estimation (lower limit) of the conductivities.

On page 6 in revised supplementary information, Table S2 and the corresponding references have been added.

Table R3. The test results for conductivity in AC mode and DC mode of the material.

	Resistance (R, Ω)	Resistivity (ρ , $\Omega\cdot\text{m}$)	Conductivity (σ , $\text{S}\cdot\text{m}^{-1}$)	Tablet diameter (ϕ , mm)	Tablet thickness (L, mm)
AC mode	5.09×10^6	5.95×10^4	1.68×10^{-5}	5.00	1.68
	4.02×10^6	4.70×10^4	2.13×10^{-5}	5.00	1.68
	4.32×10^6	5.05×10^4	1.98×10^{-5}	5.00	1.68
	3.76×10^6	4.40×10^4	2.27×10^{-5}	5.00	1.68
	4.83×10^6	5.64×10^4	1.77×10^{-5}	5.00	1.68
DC mode	2.49×10^7	2.93×10^5	3.42×10^{-6}	5.00	1.68
	2.40×10^7	2.80×10^5	3.57×10^{-6}	5.00	1.68
	2.42×10^7	2.83×10^5	3.54×10^{-6}	5.00	1.68
	2.51×10^7	2.94×10^5	3.40×10^{-6}	5.00	1.68
	2.65×10^7	3.10×10^5	3.23×10^{-6}	5.00	1.68

On page 22 in revised supplementary information, the following new reference has been added.

[1] R. A. Huggins, *Ionics* **2002**, 8, 300–313.

Question 3: What are the rationales for choosing M = O, F, S, N?

Response: This is a very interesting question. The most obvious reason is that they are light elements that are among the first to be considered when designing battery electrodes (C is also light but it is not compatible with our synthesis possibly due to the high stability of carbide). The second reason for choosing these elements is that Li₂O, LiF, Li₂S and Li₃N are all known ionic conductors, and their combination with an electronic conductor (e.g. Fe⁰ in this work) to form interfaces potentially capable of operating the space charge storage mechanism has been demonstrated for lithium storage (besides the two references mentioned by **Reviewer #1**, also *Nat. Energy* **2017**, 2, 16208 and our previous work *Nat. Mater.* **2021**, 20, 76–83). Furthermore, from the viewpoint of the materials accessibility, the compositions of these materials are simple (consisting of only two elements) and they are relatively easy to synthesize. The raw materials for the lithium thermal displacement reaction, besides the lithium, involve only iron oxide, iron fluoride, iron sulfide, and iron nitride. These compounds are all commercially available or can be facilely prepared in lab. This feature of course means a higher potential toward practical application. Finally, Li₂O, LiF, Li₂S and Li₃N readily form ultrafine nanoparticles during the lithium thermal displacement reaction process (*Nat. Energy* **2016**, 1, 15008;

Nano Lett. **2020**, 20, 546–552 and *Adv. Energy Mater.* **2016**, 6, 1600154), enabling construction of abundant electronic/ionic conductor interfaces desired in the space charge storage mechanism. Taking all these factors into consideration, the rationales for choosing M = O, F, S, N are perfectly reasonable and easily justified.

Question 4: When the authors described in the introduction: "Thermodynamic fitting and magnetic analysis, combined with structural characterization, prove the presence of only limited conversion reaction due to reactant spatial distribution confinement even at low rates", how low the C-rate threshold is?

Response: We thank the reviewer for pointing out this tiny vagueness. In this work, we conducted electrochemical performance tests of the material at current densities from 10 C to 500 C, where 10 C represents a low rate compared to the high current density of 500 C. Thus, the phenomenon of limited conversion reaction due to reactant spatial distribution confinement at relatively low rates refers to 10 C (defined based on the dominant role of space charge storage capacity), which is approximately 1 A g⁻¹. As we answered in **Question 6** from **Reviewer #3**, from the viewpoint of specific capacity, at 1 C a similar capacity with that at 10 C was measured and the spatial confinement effect also seems to work. However, as we only performed analyses at as low as 10 C, we refrain from using a lower C rate here. Further lowering the C rate will unavoidably cause substantial conversion reaction and the space charge storage mechanism will no longer apply. Please also refer to our response to **Question 3** from **Reviewer #3** for more detailed explanation.

Changes made:

On page 4 in the revised manuscript, the corresponding sentence has been revised and added: "Thermodynamic fitting and magnetic analysis, combined with structural characterization, prove the presence of only limited conversion reaction due to reactant spatial distribution confinement even at low rates (10 C), which otherwise will overshadow the desired space charge storage mechanism."

Question 5: Although LiF is a very ionic and electronic insulating materials, the authors have achieved superior cycling performance (Figure 6a and 6d). The authors need to comment on how the superior cycling was achieved based on an insulating nature of LiF. Was the Li storage mechanism in Fe/LiF the same as Fe/Li₂O cathode? Furthermore, as the authors use MIEC concept, the authors need to explain why high carbon content (20%) was used.

Response: We thank the reviewer for pointing out these issues related to conductivities. It is true that single crystalline LiF possesses an ionic conductivity ranging only from 10⁻¹¹~10⁻¹² S m⁻¹, significantly lower than that of Li₂O (10⁻⁷ S m⁻¹), indicating that it cannot independently serve as a good storage and transport carrier for Li⁺ (*ACS Nano* **2024**, 18, 1969–1981). However, one should keep in mind that these conductivities are bulk values and in the space charge storage mechanism what matters is the conductivity at the interface. It has been demonstrated in various related studies that rapid diffusion of Li⁺ can occur at the LiF interface (*Nat. Commun.* **2015**, 6, 6668; *Nano Lett.* **2016**, 16, 1497–1501; *Nat. Energy* **2017**, 2, 16208 and *Nat Energy* **2023**, 8, 340–350). In the Fe/LiF system, Li⁺ and electrons are stored on opposite sides of the heterojunction interface, without entering the bulk phases of Fe and LiF. This is the essence of the interfacial storage phenomenon, which has also been experimentally

confirmed in the literature (*Nature* **2016**, 536, 159–164; *Nat. Energy* **2018**, 3, 102–108 and *Nat. Mater.* **2021**, 20, 76–83). Our measured ionic conductivity on Fe/Li₂O (Table R3) being much higher than the bulk value also seems to prove the existence of higher than usual conductivity originated from the interface. As such an interfacial storage is also physical in nature, therefore, it is not surprising to see the superior cycling performance achieved in the Fe/LiF MIEC. Furthermore, based on the similarly outstanding performance of the Fe/LiF (as well as the Fe/Li₂S and Li₃N) when compared to Fe/Li₂O, we can conclude that the lithium storage mechanisms in these materials are the same, all relying on the space charge storage mechanism at the electronic/ionic conductor interface.

Regarding the issue of high carbon content, although Li⁺ can undergo rapid diffusion at the interfaces of relevant ionic conductors and in the liquid electrolyte throughout the whole electrode, the addition of sufficient conductive additives could enable good electronic contact between individual nanoparticles, ensuring participation of all active materials in the lithium extraction process and enhancing the lithium extraction capability of the nanoparticles (*J. Electrochem. Soc.* **2004**, 151, A1878). Otherwise, the voids among the nanoparticles will be rather detrimental to the electron conduction in the electrode and will hamper the high rate performance. Therefore, it is a common practice to apply an appropriate amount of carbon as a conductive additive even though the active material itself is highly conductive. Not to mention that the measured electronic conductivity of our Fe/Li₂O material is $3.43 \times 10^{-6} \text{ S m}^{-1}$ (Table R3), which does not exhibit significant advantages compared to traditionally conductive electrode materials. Thus adding a certain amount of carbon as a conductive additive in our case was very reasonable and we did not study the influence of different amounts of carbon on the electrode performance. The choice of 20% was based on common values reported in the literature.

Question 6: The authors can explain in detail how MIEC was incorporated in the Fe/Li₂M materials, illustrating the ionic conduction and electronic conduction contributors in the electrode.

Response: We would like to take this opportunity to explain further how MIEC was achieved in our material system. In order to form a MIEC, one can either perform physical mixing of the ionic conducting material and the electronic conducting material (e.g. layer-by-layer deposition) or carry out chemical reaction of appropriate precursors to form the ionic/electronic conducting materials (e.g. electrochemical reduction). For the purpose of maximizing the interface in the MIEC, we take the latter synthetic approach by using iron compounds (FeM_n) as the initial raw materials and reacting through a lithium thermal displacement reaction. After the reaction, the raw materials are reduced to Fe/Li₂M with metallic Fe as the electronic conductor and Li₂M as the ionic conductor. Just like the electrochemical reduction, the two form a vast amount of electronic/ionic conductor interfaces without the need for further processing. In electrochemical cycles, for example during discharge, electrons are transported and stored on the Fe side (d bands) and Li⁺ on the Li₂M side (vacancies) of the interface according to the interfacial storage mechanism (*Angew. Chem., Int. Ed.* **2013**, 52, 4998–5026). In this way, the construction and working mechanism of the MIEC have been clearly illustrated.

Changes made:

On page 5 in the revised main text, the following sentences have been added: “In normal bulk storage mechanism a material simultaneously accommodates ions and electrons and must sustain both high electronic and ionic conductivities in order to achieve a high power density (Fig. 1a). While in space

charge storage mechanism a material combines different phases that separately store and transport ions and electrons in its individual space charge zones (Fig. 1b). In this context, fabricating a mixed ionic-electronic conducting material is the key to realize the space charge storage mechanism.”

Response to Reviewer #3:

Reviewer #3: In this work, ultra-fast charging/discharging, ultra-stable and high energy charge storage properties can be realized in an artificial electrode made from a mixed electronic/ionic 2D conductor material ($\text{Fe}/\text{Li}_x\text{M}$, where $\text{M}=\text{O}, \text{F}, \text{S}, \text{N}$) enabled by a space charge principle. As a result, the $\text{Fe}/\text{Li}_2\text{O}$ electrode is able to be charged/discharged to 126 mAh g^{-1} in 6 s at a high rate of up to 50 A g^{-1} . The authors claimed that the space charge storage mechanism will play a critical role in advancing electrochemical energy storage and provides a unique perspective for designing high-performance anode materials for lithium-ion batteries. However, some problems existed in the manuscript.

Response: Thank you for carefully examining our manuscript and providing very detailed comments for us to revise and improve it. All your raised questions have been carefully answered and new results and discussions pertaining to your comments have been included in the revised manuscript.

Question 1. In 2017, Yingzhu Jiang group designed $\text{SnO}_2\text{-Fe}_2\text{O}_3\text{-Li}_2\text{O}$ nanocomposite (*Advanced Materials*, **2017**, 29, 1606499), and the high fraction of interfaces of $\text{Fe}/\text{Sn}/\text{Li}_2\text{O}$ during the charge/discharge processes ensure a reversible capacity of 350 mAh g^{-1} can be maintained at an ultrahigh current density of 80 A g^{-1} . In Jiang's work, the combination of pseudocapacitive lithium storage and spatially confined electrochemical reactions were attributed to the excellent rate performances for the Sn-based nanocomposite anode materials. However, only space charge storage mechanism is demonstrated for the enhanced rate performances, in this work.

Response: We thank the reviewer for pointing out the work by Jiang et al. on ultrafast lithium storage in a Sn-based nanocomposite anode. While acknowledge the superior performance of their $\text{SnO}_2\text{-Fe}_2\text{O}_3\text{-Li}_2\text{O}$ nanocomposite electrode material, we would like to compare this work with our work and highlight the differences between the two, which lie in the materials composition and the fundamental charge storage mechanism.

From the perspective of materials compositional and structural design, Jiang's work involves the design and preparation of a triple $\text{SnO}_2\text{-Fe}_2\text{O}_3\text{-Li}_2\text{O}$ nanocomposite using pulsed spray evaporation chemical vapor deposition. The original materials consist of a mixture of oxides, and a high fraction of interfaces of $\text{Fe}/\text{Sn}/\text{Li}_2\text{O}$ forms only after a two-step electrochemical reduction. In contrast, our material is composed of Fe^0 and Li_2O nanograins directly synthesized through a lithium thermal displacement reaction, forming a mixed electronic/ionic conducting interface from the beginning. Besides, the charge storage mechanism of the $\text{Fe}/\text{Sn}/\text{Li}_2\text{O}$ material is mainly based on pseudocapacitive alloying/dealloying reactions by tailoring an appropriate voltage window. Whereas our materials primarily rely on the space charge storage mechanism, which is more physical in nature.

It is worth emphasizing that there is a fundamental difference between the space charge storage mechanism and the pseudocapacitive storage in Jiang's work. Pseudocapacitance originates from a series of rapid and reversible Faradaic redox, electro-adsorption, or intercalation processes occurring at the surface of the electrode (as shown in Fig. R4a, *Nat. Rev. Mater.* **2019**, 5, 5–19). In contrast, in materials utilizing space charge storage mechanism for energy storage, electrons and ions are stored separately on individual sides of the electronic/ionic conducting heterointerface (as shown in Fig. R4b, *Nat. Mater.* **2021**, 20, 76–83). This biphasic transport characteristic effectively harnesses the synergistic effect of fast electron and ion transport by selecting electron and ion conductors with good transport properties, thus a high power density can be achieved. Furthermore, unlike the double-layer

capacitance or surface redox reactions (pseudocapacitance) that may exist at traditional interfaces, reducing the particle size of the electron and ion transport materials in space charge storage materials can introduce a sufficient number of interfaces, thereby realizing high energy density.

As the power-law relationship analysis on the electrochemical kinetics could also be applied to our materials, in the supplementary information (Supplementary Fig. S8, or as shown in Fig. R4c-d below), we measured the b -values of the cathodic and anodic peaks of the Fe/Li₂O electrode to be 0.97 and 0.88, respectively, which indicates a typical surface-controlled electrochemical process kinetics. However, at low charge and discharge rates, the reaction mechanisms of our materials are not as straightforward as at high rates, because there exist limited conversion reactions even with the presence of the spatial distribution confinement. According to our previous research, the fitted b -value for pure space charge storage mechanism is almost equal to 1 (*Nat. Mater.* **2021**, 20, 76–83). Nonetheless, the excellent rate performance and stability of our material at high rates (up to 500 C) are still ensured by the space charge storage mechanism.

Therefore, we believe our work presents a different simpler synthetic method and a distinct charge storage mechanism compared to the work by Jiang et al.. The achieved spatially confined electrochemical reactions on cycling stability in Jiang's work is very insightful and we decide to cite this work when discussing the crucial role of spatial distribution.

Fig. R4. (a) Electrochemical charge-storage mechanisms of pseudocapacitance. (b) Electrochemical charge-storage mechanisms of and space charge storage mechanism. (c) CV curves of the Fe/Li₂O electrode at different scan rates in the voltage range of 0.01-1.3 V. (d) b values calculated by the relationship between peak current and scan rate in CV curves.

Changes made:

On page 13 in the revised main text, the corresponding sentence have been revised and added: “The effective preservation of functional interfaces resulting from spatial distribution plays a crucial role in ensuring the long-term cycling stability of materials. A similar pseudocapacitive lithium storage based on spatially confined electrochemical reactions to avoid intercluster migration upon cycling has also been demonstrated in a previous study^[53].”

[53] Y. Jiang, Y. Li, P. Zhou, Z. Lan, Y. Lu, C. Wu, M. Yan, *Adv. Mater.* **2017**, *29*, 1606499.

Question 2. The description of Figure 1a-b in line 81-82 is simple and ambiguous, and more explanations are needed.

Response: We thank the reviewer for pointing this out and please refer to our response to **Question 6** from **Reviewer #2**.

Question 3. In the charge state of 3.0 V, no characteristic peaks of FeO can be observed, which may be attributed to the low crystallinity of the charge products. However, obvious peaks of FeO can be observed in the XPS in the Figure 3a, and the ratio of FeO should be calculated. Furthermore, the remarkable formation of FeO is observed at high current density of 10 C. Undoubtedly, the ratio of FeO will be greatly increased if the current was set at 0.1 C to the charge state of 3.0 V, which can be characterized by the XPS, and the limited conversion reaction may not correct. Why did the author choose 10 C as the charge current, in this condition? Some structural and phase change information may be missing due to the slow interfacial dynamics and large overpotential under high rates.

Response: This is a great comment and we are glad that the reviewer brought it up. First, we want to emphasize that due to the confined spatial distribution only small amount of FeO is sparsely formed at the charge state of 3.0 V (Fig. 3g), and this could also explain the absence of its characteristic peaks in XRD. Second, XPS is a surface sensitive technique and its quantitative analyses may not reflect the true ratio in the overall composition. In accordance with the reviewer's suggestion, we supplemented the fitting ratio of the FeO based on Fig. 3a. The fitting results indicate that the fraction of FeO relative to total Fe is only 26.2 %, accounting for just 16.94 % of the overall material mass (calculated based on the initial Fe elemental mass fraction of 40 %). From the fitting results, combined with the fact that XPS would overestimate the bulk situation, it is evident that the content of FeO is indeed significantly lower than that of the Fe⁰.

As this study aims to provide insights into designing high-power anode materials for lithium-ion batteries, we restricted our rate performance tests from 10 C to 500 C in the initial manuscript. As we answered in **Question 6** from the same reviewer, at 1 C the electrode is still able to operate on the space charge storage mechanism judged by the measured specific capacity. As the reviewer correctly pointed out, further lowering the C rate will eventually activate the undesired conversion reaction and most likely the high rate performance and the cycling stability will be sacrificed (as shown in our response to **Question 7** below). Therefore, we intentionally chose relatively high C rates to prevent substantial conversion reaction. The slow interfacial dynamics mentioned by the reviewer exactly refers to the sluggish charge transfer processes in the conversion reaction and this is what we want to avoid as much as possible in order to sustain the space charge storage mechanism.

Question 4. According to proposed lithium storage mechanism, FeO was the undesirable product in the space charge storage mechanism, and the readers may wonder that whether the FeO will accumulate, and therefore the amount of FeO after intensive cycles should be distinguished.

Response: This comment is actually related to the previous one and the stability of this confined FeO is certainly important to maintain the desired space charge storage mechanism. As correctly pointed out by the reviewer, FeO is indeed an undesirable product from a conversion reaction that occurs inside the voltage window (0.01-3 V) during the charging and discharging process. The comment to check whether there is accumulation of FeO after intensive cycling is definitely meaningful. If the conversion occurs only at the boundaries between the aggregated regions between the ionic conductor and electronic conductor, and if the reaction exhibits sufficient reversibility, FeO should not accumulate after cycling. To verify this, we have supplemented our study with XPS spectra of the Fe/Li₂O material after 200 cycles at 10 C, as shown in Fig. R5. Using the estimation method suggested by the reviewer in **Question 3**, we found the fitting result of the FeO ratio is similar to the fitting value before 200 cycles at 3 V (calculated based on XPS peak areas, the ratio of FeO to total Fe is approximately 30 %, versus the initial value of 26.2 %), indicating that there is no significant accumulation of FeO during cycling. This also demonstrates the high reversibility of the limited conversion reaction under the reactant spatial distribution confinement.

Fig. R5. High-resolution XPS spectra of the Fe element in the Fe/Li₂O electrode at 3 V after 200 cycles at 10 C.

Question 5. In the Figure 6a and Figure 2c, obvious pseudocapacitive lithium storage is observed, which ensure the excellent rate performances. The particle size of the Fe/Li₂O before cycling and after cycling could be added.

Response: Indeed, our demonstrated space charge storage mechanism in this work shares many similar characteristics with the well-established pseudocapacitive storage and both mechanisms imply excellent rate performance. As in the case of Fe/Li₂O charge storage mainly occurs at the interface between the two components, we expect to see no significant change of the particle size. While the particle size of Fe could be more or less accurately determined by measuring its M-H curve and performing Langevin fitting, we could only estimate that of Li₂O from microscopic imaging. For the particle size of Fe in the initial Fe/Li₂O electrode material, please refer to the M-H curve in Fig. 1f (or as shown in Fig. R6a) and the corresponding Langevin fitting results. The best-fitting particle radius of Fe is R=4.11 nm. The particle size of Li₂O in the initial Fe/Li₂O electrode material could be observed

from the HRTEM image in Fig. 3e (or as shown in Fig. R6b) and the value is approximately 4.22 nm.

In order to find the particle size of Fe after cycling, we acquired the M-H curve of the Fe/Li₂O electrode and the corresponding Langevin fitting curves after cycling for 200 cycles at a current density of 10 C, as shown in Fig. R7a. The Langevin-fitting gives a particle radius of R=3.88 nm and the saturation magnetization intensity is 89.5 emu g⁻¹ after 200 cycles (compared to fitted Fe particle size before cycling R=4.11 nm and the saturation magnetization intensity 88.8 emu g⁻¹). From Fig. R7b, it can be observed that the average particle size of Li₂O in the electrode is approximately 4.90 nm after 200 cycles (compared to an average particle size of about 4.22 nm before cycling). From these results, it can be concluded that the particle size of Fe and Li₂O after cycling did not show significant changes compared to the values before cycling. This also indicates that the material can maintain good stability during the electrochemical cycling process.

Fig. R6. (a) M-H curve of the Fe/Li₂O in initial state and the corresponding Langevin fitting curve. (b) HRTEM images of the Fe/Li₂O in initial state. Scale bar is 5 nm.

Changes made:

On page 11-12 in the revised main text, the following discussions have been added: “Furthermore, in order to check the stability of Fe and Li₂O in terms of their particle size after cycling, we first acquired the M-H curve of the Fe/Li₂O electrode and the corresponding Langevin fitting result after cycling for 200 cycles at a current density of 10 C (Supplementary Fig. S5a). The Langevin fitting gives a particle radius of R=3.88 nm and the saturation magnetization intensity of 89.5 emu g⁻¹ after 200 cycles (compared to the initial particle size of R=4.11 nm and the saturation magnetization intensity of 88.8 emu g⁻¹ before cycling, as shown in Fig.1f). Next from the HRTEM image (Supplementary Fig. S5b), it can be observed that the average particle size of Li₂O in the electrode is approximately 4.90 nm after 200 cycles (compared to an average particle size of about 4.22 nm before cycling, as shown in Fig.3e). From these results, it can be concluded that the particle sizes of Fe and Li₂O after cycling exhibit no significant changes compared to those before cycling. All above analyses confirm an electrochemically stable electrode achieved in the chemically synthesized Fe/Li₂O composite.”

On page 13 in revised supporting information, Supplementary Fig. S5 has been added as shown below.

Fig. R7. (a) M-H curve of the Fe/Li₂O at room temperature and the corresponding Langevin fitting curve after 200 cycles at a current density of 10 C. (b) HRTEM image of the Fe/Li₂O when charged to 3 V after cycling for 200 cycles at a current density of 10 C. Scale bar is 5 nm.

Question 6. The cycle performance and rate performance were obtained at high current densities, and what about the electrochemical performances of the Fe/Li₂O at small current density such as 1 C? Besides, what about the loading amount of Fe/Li₂O? Did the authors evaluate the electrochemical performances in the full cell, which is helpful for its practical applications. The comparison in Figure 2f needs to be considered with similar loading amount.

Response: In response to the reviewer's comments, we have supplemented the electrochemical performance of Fe/Li₂O at 1 C, as shown in Fig.R8. It can be observed that the Fe/Li₂O delivers an average specific capacity of ~455 mAh g⁻¹ in the first 200 cycles at 1 C. Compared with the electrochemical performance at 10 C (407 mAh g⁻¹, Supplementary Fig. S6b, the specific capacity difference is not significant. This further indicates that conversion reaction is limited due to the reactant spatial distribution confinement even at this low rate.

As regards to the mass loading and full cell performance, please refer to our responses to **Question 1** from **Reviewer #2**.

For the comparison in Fig. 2f, we have summarized the mass loading data for all selected materials in Table R4. As our electrode material has a loading amount quite comparable to the numbers in the table, we believe the comparison we made in Fig. 2f is a fair one.

Fig. R8. Cycling curves of the Fe/Li₂O electrode at the current density of 1 C for the first 200 cycles.

Table R4. The loading of active material of selected anode materials for lithium-ion batteries in the research field of fast charging.

Anode materials	Mass loadings	References
lithium yttrium titanate (LYTO)	1.5 mg cm ⁻²	Adv. Energy Mater. 2022 , 12, 2200922.
Rock salt Li ₃ V ₂ O ₅	2~3 mg cm ⁻²	Nature 2020 , 585, 63–67.
TiO ₂ /3D Cu-GM	0.537 mg cm ⁻²	ACS Nano 2022 , 16, 9762–9771.
NiNb ₂ O ₆	1~3 mg cm ⁻²	Adv. Energy Mater. 2021 , 12, 2102972.
Li ₄ Ti ₅ O ₁₂ (LTO-600)	2.0 mg cm ⁻²	Adv. Energy Mater. 2022 , 12, 2201130.

Changes made:

On page 8 in the revised main text, the following sentence has been added: “Fig. 2f further compares the rate performance with previously reported fast-charging electrodes using comparable mass loadings (Supplementary Table S4)^[41–45].”

On page 8 in revised supplementary information, Supplementary Table S4 has been added as shown below.

Table R5 The loading of active material for selected anode materials for lithium-ion batteries in the research field of fast charging.

Anode materials	Mass loadings	Reference
lithium yttrium titanate (LYTO)	1.5 mg cm ⁻²	[5]
Rock salt Li ₃ V ₂ O ₅	2~3 mg cm ⁻²	[6]
TiO ₂ /3D Cu-GM	0.537 mg cm ⁻²	[7]
NiNb ₂ O ₆	1~3 mg cm ⁻²	[8]
Li ₄ Ti ₅ O ₁₂ (LTO-600)	2.0 mg cm ⁻²	[9]

Question 7. As shown in Figure 3 and Figure 4, the authors provided a large number of data to prove the role of mixed electronic/ionic conductor interface in stabilizing the electrochemical performances, however, there is no clear data about the role of spatial distribution in stabilizing the electrochemical performances.

Response: We thank the reviewer for emphasizing the crucial influence of spatial distribution on the electrochemical performance. Actually, this points to a very subtle yet significant difference between our chemically synthesized Fe/Li₂O and conventional electrochemically reduced Fe/Li₂O. To elucidate this difference, we collected electrochemical performance of Fe₃O₄ as a comparative anode for lithium-ion batteries. It is known that during the electrochemical cycling of Fe₃O₄, elemental Fe⁰ and Li₂O nanoparticles will be also produced, but this electrochemical in-situ generation will result in a uniform distribution of Fe⁰ and Li₂O nanoparticles without specific spatial confinement (*Nat. Mater.* **2021**, 20, 76–83). The HRTEM image in Fig. 3c confirms this spatial distribution of the Fe/Li₂O system without any specific spatial confinement. Fig. R9a shows the cycling performance of the Fe₃O₄ anode for the first 300 cycles at a current density of 1 A g⁻¹, corresponding to the electrochemical performance of the Fe/Li₂O system without spatial confinement. It is evident that its cycling stability is rather poor despite a very high initial capacity, with a capacity retention of only 56 % after 300 cycles at 1 A g⁻¹, much lower than the capacity retention of the spatially confined Fe/Li₂O system at the same current density after 300 cycles (96 %, as shown in Fig. R9b). The conversion reaction between Fe and FeO is responsible for the larger capacity in the electrochemically reduced Fe/Li₂O, but it also interferes with interfacial storage mechanism and should be avoided if high rate performance is targeted. As discussed in the main text, “it can be inferred that the spatial distribution of the components in the composite electrode significantly influences the reaction depth and degree of completion for its subsequent conversion reaction. It is precisely due to a confinement effect imposed by the spatial distribution of the components that the conversion is largely limited in the chemically synthesized Fe/Li₂O composite.” Because the degree of the conversion reaction is higher when C rate is lower, therefore, we believe the cycling stability comparison shown in Fig. R9 serves as sufficient evidence to demonstrate the significant role of spatial distribution in stabilizing electrochemical performance. Relevant descriptions have been added to the manuscript.

Changes made:

On page 13 in the revised main text, the following sentences have been revised and added: “As a direct comparison, Supplementary Fig. S6a shows the cycling performance of Fe₃O₄ for the first 300 cycles at a current density of 1 A g⁻¹, corresponding to the electrochemical performance of the Fe/Li₂O system without any specific spatial confinement. It is evident that its cycling stability is poor, with a capacity retention of only 56 % after 300 cycles at 1 A g⁻¹, much lower than the capacity retention of the spatially confined Fe/Li₂O system at the same current density after 300 cycles (96 %, as shown in Supplementary Fig. S6b).”

On page 14 in revised supplementary information, Supplementary Fig. S6 has been added as shown below.

Fig. R9. (a) Cycling curves of the Fe_3O_4 at the current density of 1 A g^{-1} for the first 300 cycles. (b) Cycling curves of the $\text{Fe}/\text{Li}_2\text{O}$ electrode at the current density of 10 C for the first 300 cycles.

Question 8. What about the initial coulombic efficiency at low current density? What about the scaled production of the $\text{Fe}/\text{Li}_2\text{O}$ anode?

Response: We thank the reviewer for these two questions. The galvanostatic charge/discharge profiles for the first three cycles of the $\text{Fe}/\text{Li}_2\text{O}$ anode at 1 C are shown in Fig. R10. The initial coulombic efficiency was deviated from 100% because the initial open circuit voltage is too low (about 1.0 V , lower than the fully charged voltage of 3 V). Starting right from the second cycle, the coulombic efficiency quickly reaches a value of $\sim 100 \%$, indicating good electrochemical reversibility.

As for the question regarding scaled production of the $\text{Fe}/\text{Li}_2\text{O}$ anode or $\text{Fe}/\text{Li}_2\text{M}$ in general, the employed lithium thermal displacement reaction described in the Methods section should be easily scaled up. Due to the stringent requirements on the isolation from air or moisture, we were currently limited to carry out this reaction to achieve laboratory-scale production (gram level) inside a glovebox. Moreover, the product $\text{Fe}/\text{Li}_2\text{M}$ materials need also be stored in the glove box otherwise they would undergo rapid oxidation or corrosion. We certainly believe that future research to tackle more suitable experimental conditions and process optimization can solve this engineering problem, but this would not take away the key insights and important learnings demonstrated in our current study.

Fig. R10. The first three cycling steps of Fe/Li₂O for lithium ions storage at 1 C.

REVIEWERS' COMMENTS

Reviewer #1 (Remarks to the Author):

The manuscript is well revised and can be published in the current form.

Reviewer #2 (Remarks to the Author):

The authors have resolved all of my comments. Thus, I suggest this manuscript being a publication in Nature Communications.

Reviewer #3 (Remarks to the Author):

All the questions have been well addresses by a large number of additional experiments, and it can be accepted without further revisions.